# Universal inter-molecular radical transfer reactions on metal surfaces

Junbo Wang[1,2,7], Kaifeng Niu[2,3,7], Huaming Zhu[1,7], Chaojie Xu[2], Chuan Deng[1], Wenchao Zhao[4], Peipei Huang[1], Haiping Lin [1], Dengyuan Li[5], Johanna Rosen[3], Peinian Liu [5], Francesco Allegretti[4], Johannes V. Barth[4], Biao Yang [2,4] ✉, Jonas Björk [3] ✉, Qing Li [1] ✉ & Lifeng Chi [2,6] ✉

On-surface synthesis provides tools to prepare low-dimensional supramolecular structures. Traditionally, reactive radicals are a class of single-electron species, serving as exceptional electron-withdrawing groups. On metal surfaces, however, such species are affected by conduction band screening effects that may even quench their unpaired electron characteristics. As a result, radicals are expected to be less active, and reactions catalyzed by surface-stabilized radicals are rarely reported. Herein, we describe a class of inter-molecular radical transfer reactions on metal surfaces. With the assistance of aryl halide precursors, the coupling of terminal alkynes is steered from non-dehydrogenated to dehydrogenated products, resulting in alkynyl-Ag-alkynyl bonds. Dehalogenated molecules are fully passivated by detached hydrogen atoms. The reaction mechanism is unraveled by various surface-sensitive technologies and density functional theory calculations. Moreover, we reveal the universality of this mechanism on metal surfaces. Our studies enrich the on-surface synthesis toolbox and develop a pathway for producing low-dimensional organic materials.

Free radicals are single-electron species created through the homolysis of covalent bonds via external stimuli[1,2]. The presence of unpaired electrons results in their strong oxidizing properties[3]. Reactive radicals facilitate a broad range of addition, fragmentation, group-transfer, and recombination reactions[4,5]. In conventional chemistry, both inter- and intra-molecular radical transfer processes play a key role in radical reactions[6,7]. Eq. (1) illustrates a typical group-transfer radical reaction in solution[8,9]. Cl· radical reduces the C−H bond activation barrier of CH₄ considerably[10]. Accompanied by the inter-molecular radical transfer from Cl· to CH₄, the detached hydrogen atom migrates and

interacts with Cl·, forming HCl products.

$$Cl \cdot + CH_4 \rightarrow CH_3 \cdot + HCl \tag{1}$$

Conventionally, the processes of radical reactions are investigated by electron spin resonance (ESR) spectra[11]. However, the lack of real-space recognition of reaction intermediates and products at the single-molecule level entailed long-standing debates about nature of surface-stabilized radicals and the mechanism of inter-molecular radical

[1]School of Physics and Information Technology, Shaanxi Normal University, Xi'an 710119, China. [2]Institute of Functional Nano & Soft Materials (FUNSOM), Jiangsu Key Laboratory for Carbon-Based Functional Materials & Devices, Soochow University, Suzhou 215123, China. [3]Department of Physics, Chemistry and Biology, IFM, Linköping University, Linköping 58183, Sweden. [4]Physics Department E20, Technical University of Munich, James-Franck-Str. 1, 85748 Garching, Germany. [5]Key Laboratory for Advanced Materials and Feringa Nobel Prize Scientist Joint Research Center, Frontiers Science Center for Materiobiology and Dynamic Chemistry, State Key Laboratory of Chemical Engineering, School of Chemistry and Molecular Engineering, East China University of Science & Technology, Shanghai 200237, China. [6]Department of Materials Science and Engineering, Macau University of Science and Technology, Macau 999078, China. [7]These authors contributed equally: Junbo Wang, Kaifeng Niu, Huaming Zhu. ✉e-mail: biao.yang@tum.de; jonas.bjork@liu.se; liqing@snnu.edu.cn; chilf@suda.edu.cn

transfer processes. These issues are of crucial relevance for the recently established on-surface chemistry[12–15]. Through bottom-up surface synthesis strategies, diverse low-dimensional functional nanostructures with atomic precision have been successfully constructed using well-defined substrates[16–26]. Specifically, on-surface synthesis possesses the following advantages: (1) Reaction precursors, intermediates, and products can be characterized in situ down to the subatomic level with the aid of high-resolution scanning probe microscopy[27–32]. (2) On-surface chemistry provides ideal model systems for exploring innovative reaction and fabrication protocols, proceeding on atomically clean single-crystals under ultrahigh vacuum conditions[33–36]. (3) Importantly intermediate states elusive in solution reactions or hitherto unknown metastable species can be stabilized through the appreciable interactions between adsorbates and the underlying surface atomic lattice[37–39].

On-surface chemistry provides opportunity to investigate the unequivocal pathways of specific reactions at single-molecule scale. However, achieving inter-molecular radical transfer reactions on surfaces still remains challenging[23], primarily due to the following reasons: (1) Radicals on surfaces are believed to be less active since the unpaired electrons are quenched on metal surfaces[18,40,41]. (2) Using aryl halide precursors as an example, successful inter-molecular radical transfer requires that the radical transfer process precedes the self-passivation of phenyl radicals by forming Ph-Ph (on gold) or Ph-M-Ph (M = Ag, Cu) bonds[42,43]. Such spurious effects can be partly voided by performing chemical reactions on non-metal surfaces[44,45].

Herein, utilizing aryl halides and alkyne derivatives as molecular precursors, we successfully achieved inter-molecular radical transfer reactions on metal surfaces. Surface-stabilized phenyl radicals are generated by thermally activating Ph-X (X = Br and I) bond scission on Ag(111). The formed radicals facilitate dehydrogenation reactions of alkyne groups and alter reaction pathways of alkyne groups from non-dehydrogenated to dehydrogenated coupling reactions. Specifically, temperatures required to initiate the coupling of alkyne groups reduce

from 400 K (without aryl halide precursors) to 340 K when taking aryl bromide as the radical initiator, and to below room-temperature (RT) when taking aryl iodides as the radical initiator on Ag(111) (Fig. 1a, b). Similar to that chemical conversion shown in Eq. (1), accompanying the radical transfer process, detached hydrogen atoms migrate and passivate phenyl radicals with a 100% yield. The newly created alkynyl radicals interconnect via alkynyl-Ag-alkynyl bonds. In contrast, without the assistance of aryl halide precursors, alkyne derivatives engage in non-dehydrogenative coupling reactions on Ag(111) (Fig. 1a, c), as previously shown[23,24,32,46]. Moreover, we found that dehydrogenation reactions of amino/imino groups can also be triggered at much lower temperatures through the same radical transfer mechanism on Ag(111) and Cu(111), respectively, suggesting that this mechanism is of general relevance. The entire reaction pathways, evidenced by scanning tunneling microscopy (STM) on the molecular level, are confirmed by extensive density functional theory (DFT) calculations and further supported by temperature-programmed desorption (TPD) and temperature-programmed X-ray photoelectron spectroscopy (TP-XPS) experiments. Our investigations thus not only report inter-molecular radical transfer reactions on metal surfaces but also indicate their broad relevance in on-surface chemistry.

## Results and discussion
### Reaction behavior on Ag(111)
Deposition of 1,3,5-tris-(4-ethynylphenyl) benzene (Ext-TEB) on Ag(111) held at RT results in densely packed islands (Supplementary Fig. 1a, b). Coupling of Ext-TEB molecules take place after annealing at 400 K (Supplementary Fig. 1c, d)[23,24,32,46]. On the other hand, depositing 4,4″-dibromo-p-terphenyl (DBTP) on Ag(111) held at RT leads to the formation of two-dimensional porous network (Supplementary Fig. 1e, f). The initial debromination of DBTP is observed upon annealing at 370 K (Supplementary Fig. 1g, h). The temperatures needed to trigger coupling of Ext-TEB and debromination of DBTP are in excellent agreement with previous reports[24,47].

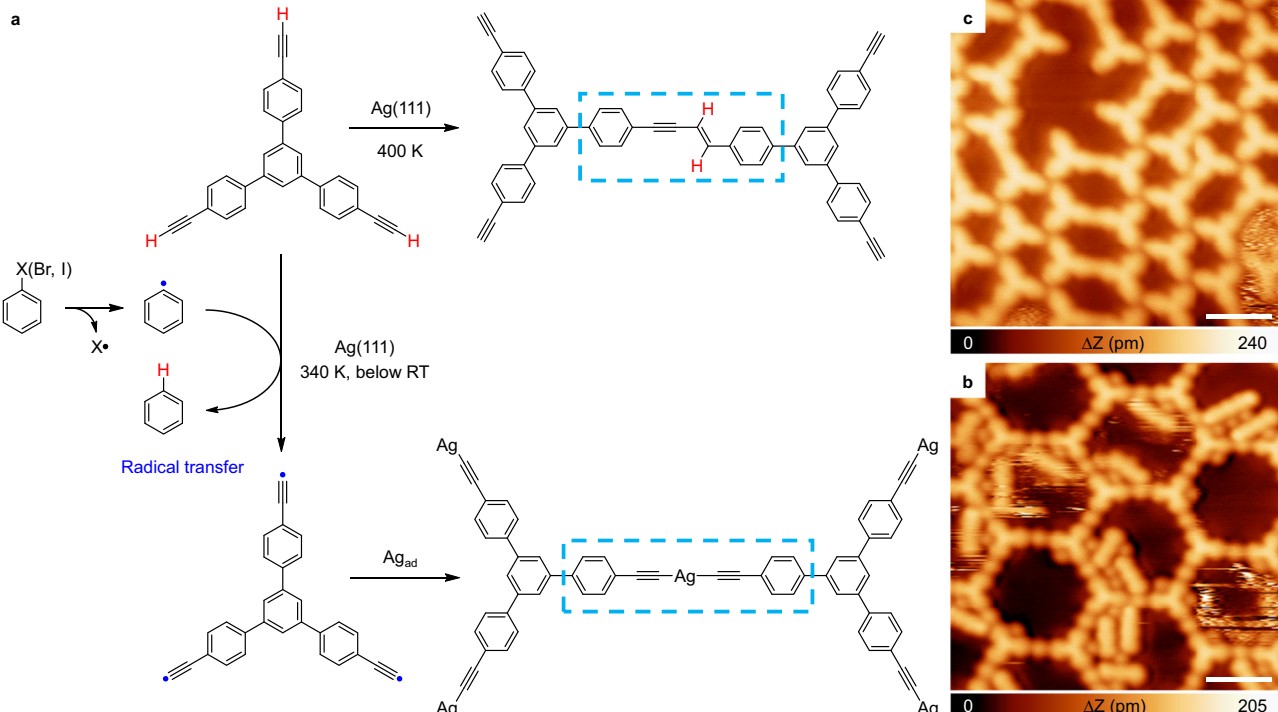

**Fig. 1 | Schematic illustration of the inter-molecular radical transfer reaction. a** Scheme of the reaction behaviors of Ext-TEB, with/without the assistance of phenyl radicals on Ag(111). **b, c** High-resolved STM images of the corresponding reaction products. Tunneling parameters are $I_t = 100$ pA and $V_b = -1$ V for all the STM images. Scale bars: 2 nm.

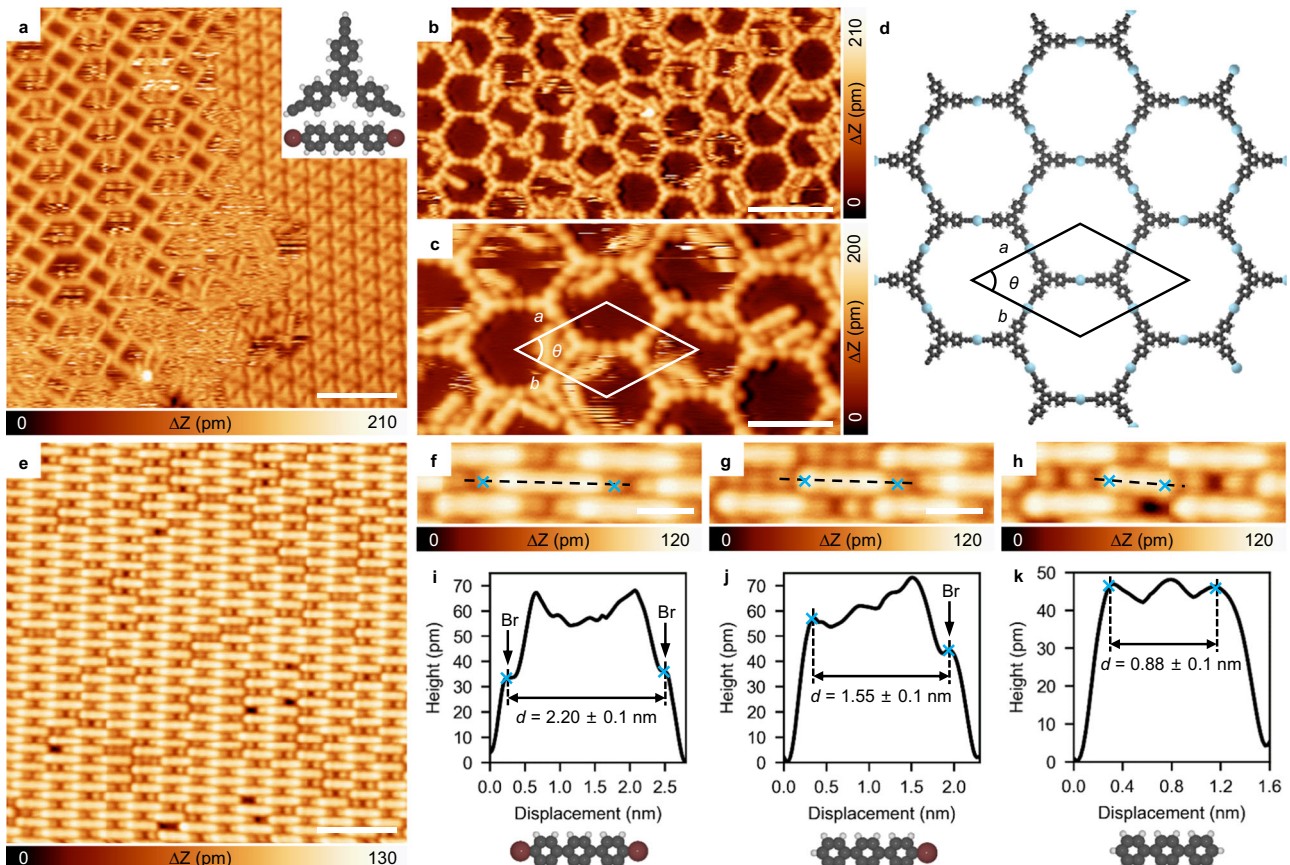

**Fig. 2 | Reaction behavior after co-depositing Ext-TEB and DBTP on Ag(111). a** STM topographic image after co-deposition of DBTP and Ext-TEB with the stoichiometric ratio 3:2 on Ag(111) held at RT. **b, c** Large-scale and zoomed-in STM images of the alkynyl-Ag network after annealing the sample at 340 K. **d** DFT optimized structural model of the alkynyl-Ag network. **e** Self-assembly of DBTP after annealing the sample at 340 K. **f–h** High-resolution STM images of three representative reaction products of DBTP. **i–k** The corresponding line profiles and structural models. Tunneling parameters are $I_t = 100$ pA and $V_b = -1$ V for all the STM images. Ag, Br, C, and H atoms are represented by blue, brown, gray, and white circles, respectively. Scale bar: **a**, **b** 6 nm, **c** 3 nm, **e** 4 nm, **f–h** 1 nm.

Figure 2a gives a representative STM image after co-depositing DBTP and Ext-TEB molecules with a stoichiometric ratio of 3:2 on Ag(111) held at RT. Precursor molecules self-assemble into separated phases, and each phase is same to that observed in Supplementary Fig. 1. Upon annealing the sample at 340 K, triangle-shaped Ext-TEB molecules connect with each other, forming porous networks, as depicted in Fig. 2b (the corresponding large-scale STM image is presented in Supplementary Fig. 2). Closer investigations (Fig. 2c) reveal that adjacent Ext-TEB monomers are linked by bright dots, implying the formation of alkynyl-Ag-alkynyl linkages[48–51], which differs from the coupling reactions of isolated alkynes on silver (Supplementary Fig. 1d)[23,24,32,46]. The optimized structural model of the porous network is shown in Fig. 2d. The calculated unit cell parameters ($a = b = 3.60$ nm, $\theta = 60°$) are in close agreement with the experimental observations ($a = b = 3.56 \pm 0.10$ nm and $\theta = 60 \pm 1°$).

The formation of Ext-TEB porous nanonetworks is accompanied by partial debromination of DBTP molecules (Fig. 2e). Figure 2f–h show three representative high-resolution STM images of linear-shaped objects. Dark protrusions are observed at both terminals of the monomer shown in Fig. 2f, whereas one or no protrusion is discernible in monomers depicted in Fig. 2g, h. According to previous reports, these protrusions can be assigned to Br atoms bonded to phenyl rings[16,52]. As Br atoms are removed, the molecules become shorter ($2.20 \pm 0.10$ nm in Figs. 2i, $1.55 \pm 0.10$ nm in Fig. 2j, and $0.88 \pm 0.10$ nm in Fig. 2k, respectively). Further detailed information on the Br abstraction is presented in Supplementary Fig. 3. Statistically, 60% of Br atoms are detached at this stage. After annealing the sample at

370 K, all Br atoms are detached (Supplementary Fig. 4). Importantly, a 100% yield of terphenyl monomers indicates that all debrominated DBTP are passivated by H atoms, rather than forming Ph-Ag-Ph bonds. The passivation of phenyl radicals by hydrogen atoms is further confirmed via control experiments (Supplementary Figs. 5 and 6). It is also noteworthy that similar phenomena are observed by performing the same experimental procedure with the ion gauge switched off, suggesting that the residue of H atoms in the chamber does not influence the reaction. After annealing the sample at 450 K, significant desorption is observed for both alkynyl-Ag species and passivated phenylenes (see Supplementary Fig. 7 for details), in agreement with the drastic decrease of the intensity of C 1$s$ spectra observed in parallel TP-XPS measurements of the (DBTP + Ext-TEB)/Ag(111) sample (Supplementary Fig. 8).

Summarizing, four distinct phenomena are encountered in the bicomponent system: (1) The temperature needed to trigger the coupling of Ext-TEB is reduced from 400 to 340 K. (2) Dehydrogenation of Ext-TEB is accompanied by debromination of DBTP. (3) Debrominated phenyl radicals are passivated by H atoms with a 100% yield. (4) The dehydrogenated alkynyl groups are interlinked via alkynyl-Ag-alkynyl bonds incorporating adatoms supplied by the substrate.

## Mechanisms of radical transfer

An earlier study already pointed out that the Ag adatom exhibits limited activity regarding the dehydrogenation of alkynes[53]. Traditionally, Ullmann-like reactions of aryl bromides on silver involve the creation of surface-stabilized phenyl radicals[18,41], followed by formation of

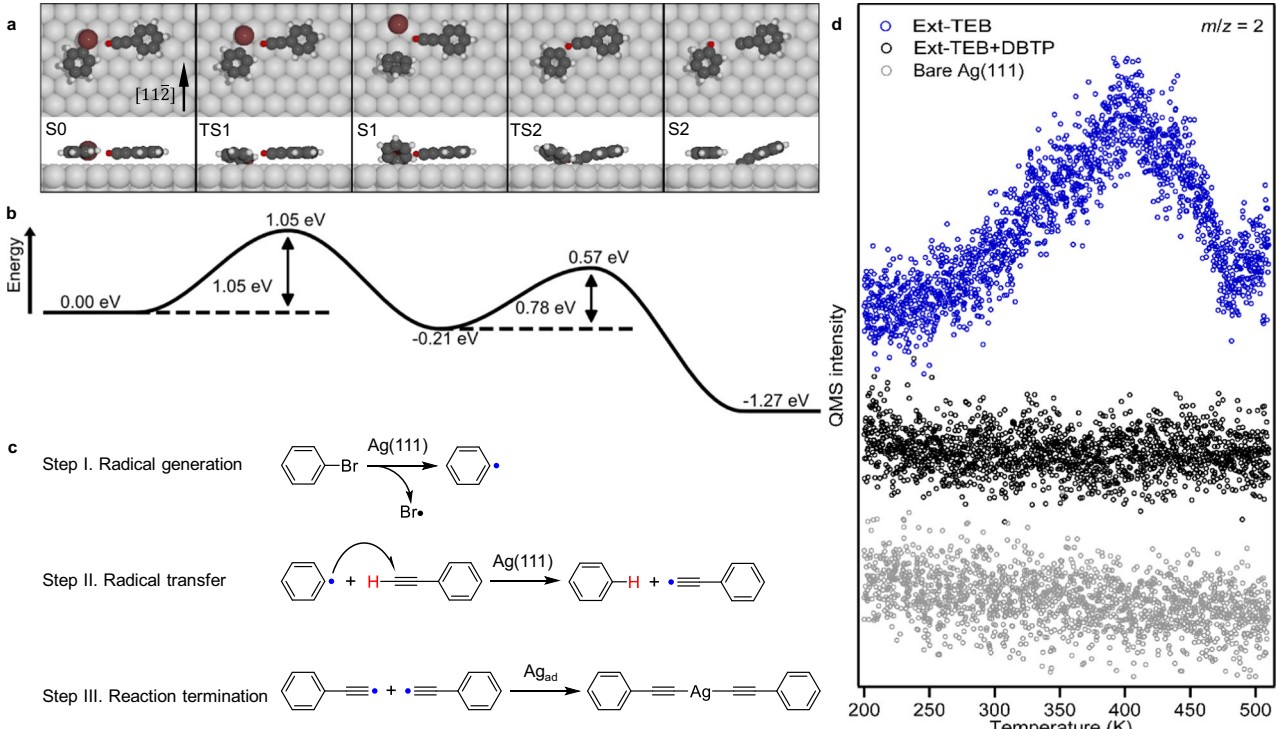

**Fig. 3 | The inter-molecular radical transfer reaction. a** DFT-calculated reaction pathway of the bi-component system (bromobenzene and phenylacetylene) on Ag(111). **b** The corresponding energy profile. The energy profile is aligned with the chemical potential of adsorbed halogen atom on Ag(111), in which the chemical potential of halogen atom $\mu_{halogen}$ is calculated as $\mu_{halogen} = E_{halogen+Ag(111)} - E_{Ag(111)}$. Ag, C, H, and Br atoms are represented by silver, gray, white, and brown circles, respectively. The H dissociated from the phenylacetylene is highlighted by red. **c** Schematic illustration of the three steps of the inter-molecular radical transfer reaction. **d** TPD spectra of hydrogen gas (H$_2$, $m/z = 2$) collected on a bare Ag(111) (gray), Ext-TEB/Ag(111) (blue), and (Ext-TEB + DBTP)/Ag(111) (black) samples with a heating rate of $1\,K\,s^{-1}$. Source data are provided as a Source Data file.

Ph-Ag-Ph intermediate states and eventually achieving Ph-Ph couplings. The aryl bromide assisted C-H bond activation of terminal alkyne is[54-56], therefore, facilitated by following candidate active centers: chemically adsorbed Br adatoms[23,57], hybrid Ag in Ph-Ag-Ph intermediates[58] and surface-stabilized phenyl radicals[23]. Br atoms/hybrid Ag species can be ruled out as active centers in our bi-component system through systematic control experiments and DFT calculations (see Supplementary Figs. 9 and 10 for details).

Accordingly, surface-stabilized phenyl radical transient state mediation is recognized as the only possible mechanism that facilitates C-H bond activations of alkynes. To illustrate detailed reaction pathways, systematic DFT calculations were carried out (Fig. 3). For simplicity, we used bromobenzene and phenylacetylene as model species. As both precursors diffuse sufficiently at 340 K on Ag(111) (see Supplementary Fig. 11 for details), the reaction initiates with the co-adsorption of bromobenzene and phenylacetylene on Ag(111), in which the terminal alkyne group heads towards the C-Br bond of bromobenzene (S0). The overall reaction process consists of two steps: debromination and radical transfer by H migration. Firstly, the phenyl radical is formed via the debromination reaction by overcoming an activation energy of 1.05 eV (Fig. 3a, step I in Fig. 3c). This barrier is similar to that without the affinity of an alkyne group (0.97 eV in Supplementary Fig. 18), implying that the presence of alkyne does not affect the debromination reaction considerably. The influence of the detached Br atom on the subsequent reaction is negligible due to the low diffusion barrier on Ag(111) (Supplementary Fig. 12). By affinity to the strong electron-withdrawing phenyl radical, phenylacetylene undergoes a dehydrogenation reaction and form an alkynyl radical with activation energy of 0.78 eV. For reaction without the assistance of aryl halide precursors, the C-C coupling[32,53,59] is energetically more favorable compared to the direct C-H activation of a terminal alkyne

group on Ag(111)[48,53]. The energy barrier of the radical transfer process is, however, lower than that for the C-C coupling of two terminal alkynes. The decreased activation energy corroborates the experimental observation that the reaction temperature reduces from 400 to 340 K upon introducing DBTP precursors on Ag(111). Importantly, the detached hydrogen atom migrates from the alkyne group to the phenyl radical, producing a passivated benzene monomer (S2 in Fig. 3a), which explains why the Ullmann-like coupling is prohibited in the bi-component system. The process from S1 to S2 can equivalently be considered as a surface-stabilized radical substitution reaction via the inter-molecular radical transfer from the phenyl to the alkynyl group (step II in Fig. 3c). The calculated overall reaction pathway is rather similar to the inter-molecular radical transfer process in solution (Eq. (1)).

The radical transfer is favored thermodynamically due to the high dissociation enthalpy of Ph-H bonds, which significantly stabilizes the final state (S2). As shown in the energy profile (Fig. 3b), the debromination of bromobenzene (from S0 to S1) is the rate-limiting step, and the radical transfer takes place spontaneously as soon as the surface-stabilized phenyl radical is created, explaining the high reaction yield. Moreover, hydrogen migration results in the generation of the alkynyl radical (S2). Two alkynyl radicals are subsequently linked by a silver adatom, forming the alkynyl-Ag-alkynyl motif (step III in Fig. 3c, the detailed process is provided in Supplementary Fig. 13), which differs from the coupling reaction of two isolated alkyne groups. It is noteworthy that a previous report claims that stabilized radicals could lead to changes of the molecular configuration, which may facilitate certain on-surface reactions[60]. In our bi-component system, the configuration of phenylacetylene is hardly affected by the chemisorption of the phenyl radical. In conclusion, our calculated reaction pathway successfully interprets the aforementioned four phenomena observed

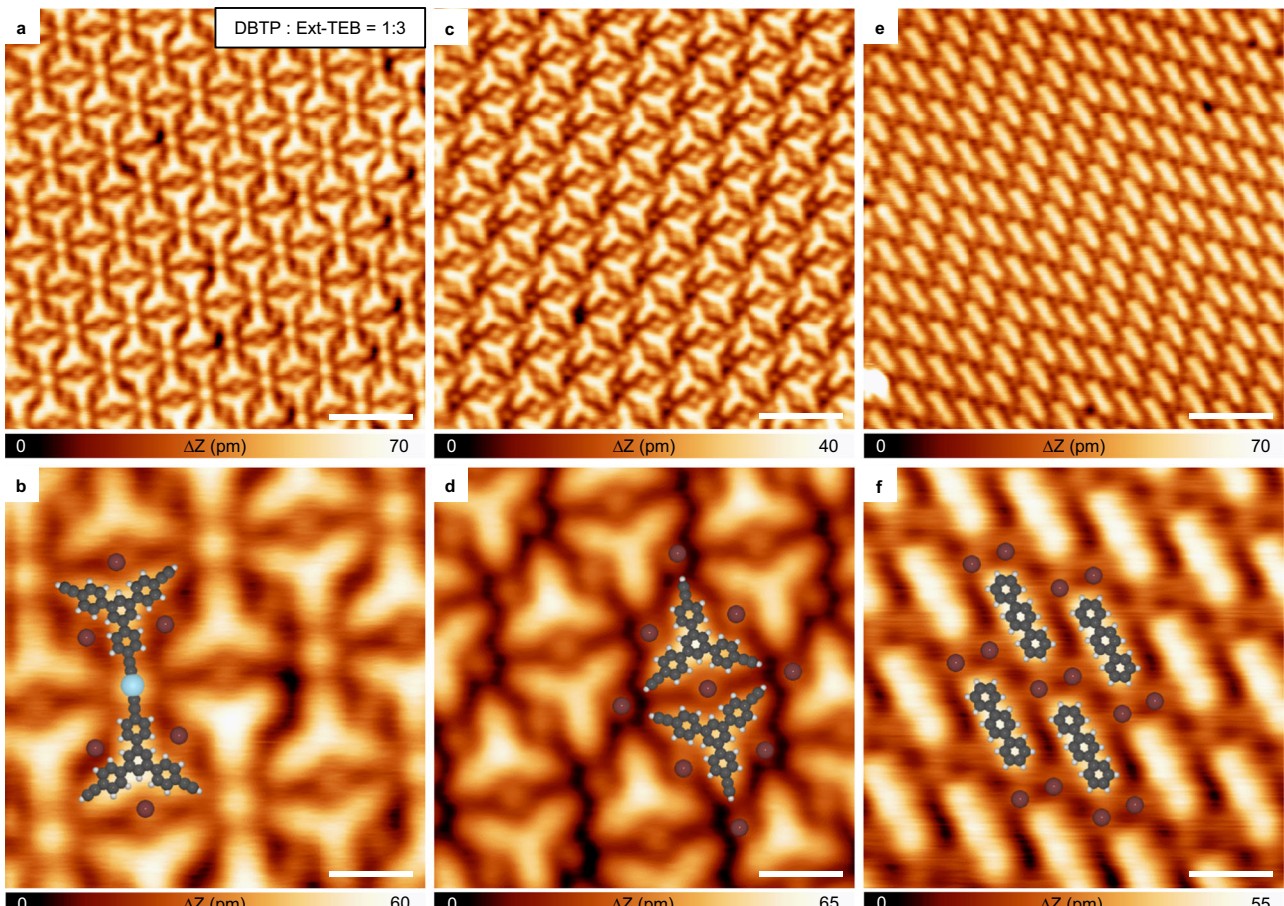

**Fig. 4 | Reaction behavior after co-depositing DBTP and Ext-TEB molecules with a stoichiometric ratio of 1:3. a, c, e** STM topographic images obtained by co-depositing DBTP and Ext-TEB onto Ag(111) with the stoichiometric ratio of 1:3, followed by annealing the sample to 370 K. **b, d, f** The corresponding high-resolution STM images. Tunneling parameters are $I_t = 100$ pA and $V_b = -1$ V for all the STM images. Ag, Br, C, and H atoms are represented by silver, brown, gray, and white circles, respectively. Scale bar: **a, c, e** 3 nm, **b, d, f** 1 nm.

experimentally, i.e. the temperature needed to trigger the coupling reaction of Ext-TEB decreased markedly; dehydrogenation always accompanies the debromination reaction; debrominated molecules are passivated by hydrogen atoms with a 100% yield; the formation of alkynyl-Ag-alkynyl bonds.

To further corroborate the suggested mechanism, TPD experiments were carried out. The background concentration of hydrogen atoms in the chamber was negligible, as no apparent $H_2$ signal was observable on bare Ag(111) (Fig. 3d). After depositing Ext-TEB solely on Ag(111), a pronounced $H_2$ signal appeared at a temperature of 400 K in the TPD trace of $m/z = 2$ (blue line in Fig. 3d). This phenomenon probably arises from that certain side reactions are suppressed by the radical transfer reactions. On the other hand, with the co-deposition of Ext-TEB and DBTP on Ag(111), no $H_2$ signal is observed in the temperature range from 200 to 500 K (black line in Fig. 3d), as all detached H atoms participate in the passivation of phenyl radicals.

Control experiments were conducted to investigate the influence of the stoichiometric ratio of two precursor molecules. When introducing an excess of Ext-TEB in the bi-component system (the stoichiometric ratio of DBTP and Ext-TEB is 1:3), only a portion (approximately 20%) of alkyne groups undergo C-H bond activation after annealing at 370 K, as shown in Fig. 4a, b. Remaining terminal alkynes maintain pristine (Fig. 4c, d), as the temperature required to trigger the coupling of Ext-TEB without the assistance of an aryl halide precursor is much higher (Supplementary Fig. 1d). All DBTP monomers undergo the detachment of Br atoms, and generated phenyl radicals

are passivated by hydrogen atoms with 100% yield (Fig. 4e, f, the reason why plenty of Br adatoms are observed can be seen in Supplementary Discussion 1). On the other hand, when introducing an excess of DBTP, all dehydrogenated alkynyl groups interact with phenyl groups and adatoms, expressing phenyl-Ag-alkynyl bonds. As DBTP molecules are in excess, phenyl-Ag-phenyl motifs are also observed (see Supplementary Fig. 14 for details). The control experiment not only clarifies that the presence of phenyl radicals reduces the temperature required to initiate the coupling reaction of alkynes, but also confirms that a single phenyl radical is able to promote the detachment of one hydrogen from the alkyne group, in agreement with the calculated reaction pathway in Fig. 3.

As shown in Fig. 3a, the dehydrogenation reaction temperature of alkyne is determined by the balance of radical creation (S0–S1) and inter-molecular radical transfer (S1–S2). To further confirm the validity of the radical transfer mechanism, additional experiments were performed in which Br was substituted by other halogen atoms (I and Cl). Among the aryl halide molecules, the threshold order of abstraction temperatures obeys the relation $T_I < T_{Br} < T_{Cl}$ (as example, separated aryl iodide and aryl chloride on Ag(111) are compared in Supplementary Fig. 15). As expected, by choosing 4,4″-diiodo-p-terphenyl (DITP) and Ext-TEB as reaction precursors (structure models are shown in Fig. 5), the dehydrogenation of alkyne group takes place at a low temperature (below RT). Accompanied by the formation of alkynyl-Ag-alkynyl motifs, deiodination and hydrogen passivation reactions of DITP occur with 100% reaction selectivity (Fig. 5a, b). On the other

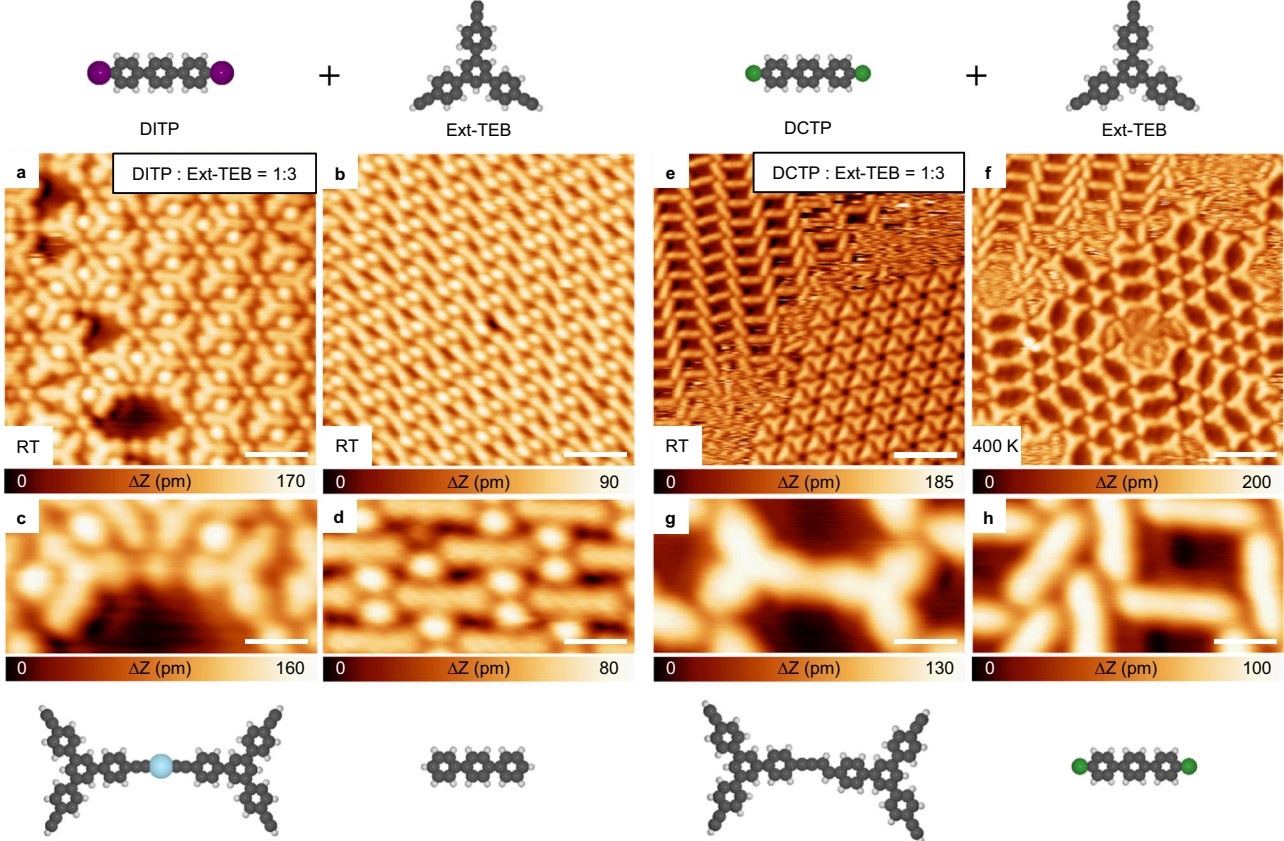

**Fig. 5 | Reaction behavior after co-depositing Ext-TEB with DITP or DCTP on Ag(111). a, b** STM images of the alkynyl-Ag network and passivated DITP island after co-depositing DITP and Ext-TEB (the stoichiometric ratio is 1:3) onto Ag(111) at RT. **c, d** Corresponding zoomed STM images and schematic structural models. **e** Assembled structures after co-depositing DCTP and Ext-TEB (the stoichiometric ratio is 1:3) on Ag(111) held at RT. **f** STM image after annealing the sample at 400 K. **g, h,** Zoomed STM images and schematic structural models. Tunneling parameters are $I_t = 100$ pA and $V_b = -1$ V for all the STM images. Ag, C, Cl, H, and I atoms are represented by blue, gray, green, white, and purple circles, respectively. Scale bar: **a, b** 2 nm, **c, d, g, h** 0.8 nm, **e, f** 4 nm.

hand, when chlorobenzene (4,4″-dichloro-p-terphenyl (DCTP)) and Ext-TEB are chosen as reactants (structural models are shown in Fig. 5), coupling of Ext-TEB takes place after annealing at 400 K, and the alkynyl-Ag-alkynyl products are not observed (Fig. 5g). Since phenyl radicals are not generated at this annealing temperature, chlorobenzene precursors remain pristine (Fig. 5h). Ullmann coupling of chlorobenzene precursors occurs after annealing at 450 K (Supplementary Fig. 16).

To elucidate the reaction pathways of Ext-TEB with different aryl halide precursors, additional DFT calculations were carried out (Fig. 6a–c). Given low diffusion barriers of halogen atoms on Ag(111) (Supplementary Fig. 12), the influence of detached halogen atoms (I, Br, and Cl) on the radical transfer process is negligible (S1–S2 in Fig. 3 and Supplementary Fig. 17). Consequently, energy barriers of radical transfer processes are identical (0.78 eV) for different aryl halide precursors. The barriers needed to overcome to create surface-stabilized phenyl radicals (S0–S1), however, vary significantly (0.71 eV for iodobenzene, 1.05 eV for bromobenzene, and 1.41 eV for chlorobenzene, respectively). As a result, the limiting energy barrier for the entire dehydrogenation reaction process is 0.78 eV for iodobenzene, 1.05 eV for bromobenzene, and 1.41 eV for chlorobenzene. Our calculations naturally explain why the dehydrogenation reaction of alkyne groups occur below RT with DITP and Ext-TEB as precursor molecules (Figs. 5a and 6d). On the other hand, the reaction temperature for dechlorination of DCTP is significantly higher than that for coupling of isolated Ext-TEB, which inhibits inter-molecular radical transfer reactions. Consequently, the formation of alkynyl-Ag-alkynyl products is prohibited (Figs. 5g and 6f)[23,24,32,46].

Aside from the reduced dehydrogenation temperature of Ext-TEB in the bi-component system, the debromination temperature of DBTP reduces from 370 K to 340 K. However, the systematic monitoring of the Br 3d signature using TP-XPS with different heating rates revealed no distinct difference in the kinetics of debromination reaction between the isolated DBTP system and the bi-component system (Supplementary Figs. 18 and 19). This result is corroborated by DFT calculations, indicating that the affinity of phenylacetylene to bromobenzene does not reduce the debromination barrier (Supplementary Fig. 18e, f). The observed 30 K temperature difference may reflect that the newly formed C-H bond of the passivated DBTP is more stable than the Ph-Ag bond (Detailed explanations are provided in Supplementary Note 17).

## Universality of radical transfer

As shown in Fig. 3, the radical transfer from S1 to S2 is favored energetically since the bond dissociation enthalpy of the Ph–H bond exceeds that of the alkynyl-H bond. Therefore, it is intriguing to investigate the universality of surface-stabilized phenyl radical-induced dehydrogenation reactions on metal surfaces. We systematically studied the reaction behaviors of various amino/imino derivatives with/without the presence of aryl bromide precursors on Ag(111) and Cu(111), respectively[61]. For example, depositing 4-amino-p-terphenyl (APTP) on Cu(111) leads to fuzzy STM patterns, suggesting pristine APTP monomers diffuse quickly on Cu(111) at 77 K. The reaction of APTP takes place upon annealing at 370 K. In contrast, depositing 4′-bromo-1,1′-biphenyl-4-amine (BBPA) on Cu(111) at RT stimulates dehydrogenation reactions of amino groups, accompanied

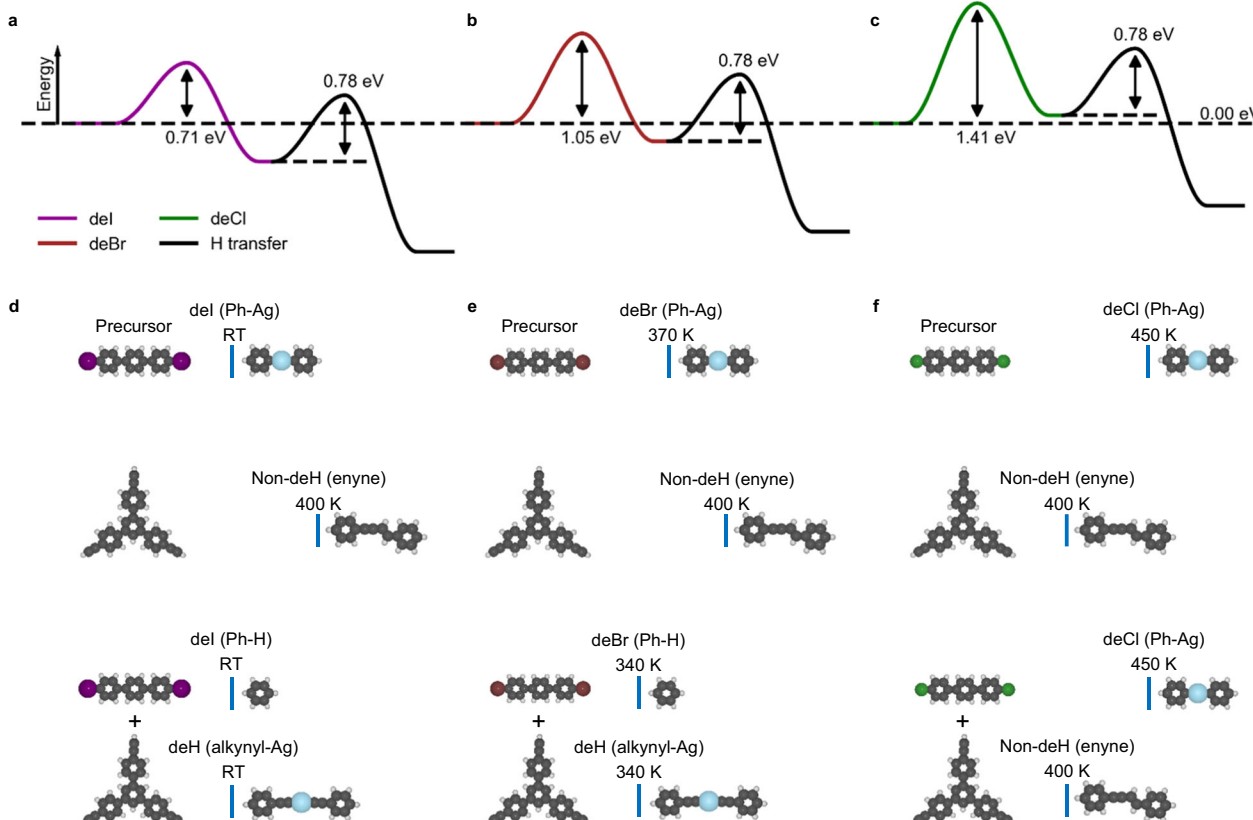

**Fig. 6 | The reaction mechanism after depositing Ext-TEB with different aryl halide precursors. a–c** The energy profiles for the radical creation (purple curve for deiodination, brown curve for debromination, and green curve for dechlorination, respectively) and inter-molecular radical transfer from phenyl to alkynyl (black curve). **d–f** The temperatures for dehalogenation, non-dehydrogenative and dehydrogenative coupling of halobenzene/Ag(111), Ext-TEB/Ag(111), and (halobenzene + Ext-TEB)/Ag(111), respectively. Ag, Br, C, Cl, H, and I atoms are represented by blue, brown, gray, green, white, and purple circles, respectively.

**Table 1 | The universality of the inter-molecular radical transfer reaction**

| Substrates | Cu(111) | Ag(111) | Ag(111) | Cu(111) | Ag(111) |
|---|---|---|---|---|---|
| Precursor molecules (without aryl bromide) | APTP | | DPB | DAQP | |
| Dehydrogenation temperature | 370 K | Desorb before dehydrogenation | Desorb before dehydrogenation | 340 K | Desorb before dehydrogenation |
| Precursor molecules (with aryl bromide) | BBPA | | BBPB | DAQP + TBB | |
| Dehydrogenation temperature | Below RT | 340 K | 390 K | Below RT | 340 K |

APTP, BBPA, DPB, BBPB, DAQP, and TBB refer to 4-amino-p-terphenyl, 4'-bromo-1,1'-biphenyl-4-amine, N,N'-diphenylbenzidine, N,N'-bis(4-bromophenyl) benzidine, 4,4"-diamino-p-quaterphenyl, and 1,3,5-tris(4-bromophenyl) benzene, respectively.

by the hydrogen passivation of debrominated phenyl radicals. The details of the control experiment are clarified in Supplementary Fig. 20. Similarly, with the presence of aryl bromides, the temperature required to trigger dehydrogenations of amino/imino groups reduces significantly for other selected precursor molecules (Table 1). Importantly, accompanied with the dehydrogenation of amino/imino groups, detached hydrogen atoms always migrate and passivate the debrominated phenyl radicals with a 100% yield. We calculated the corresponding reaction pathways of phenylamine with/without the assistance of surface-stabilized phenyl radical on Ag(111) and Cu(111), respectively. As shown in Supplementary Fig. 21, the entire radical transfer process is an exothermic reaction. With the introduction of the phenyl radical, the energy barrier of dehydrogenation of phenylamine reduces dramatically from 1.98 to 0.86 eV on Ag(111), and from 1.45 eV to 0.99 eV on Cu(111), respectively.

Our systematic experiments and theoretical calculations demonstrate the universality of phenyl radical-triggered dehydrogenation reactions on metal surfaces. Additionally, there are exciting opportunities for further exploration: For instance, we can investigate the possibility of achieving inter-molecular radical transfer reactions with different types of radicals. Moreover, exploring methods to postpone the termination of the radical reactions opens the potential for realizing well-known radical chain reactions in conventional chemistry on metal surfaces.

In conclusion, we identified a class of inter-molecular radical transfer reactions on metal surfaces. The strong electron-withdrawing phenyl radicals, generated from homolysis of Ph-X bonds (X = Br, I), facilitate the C-H activation of alkyne groups with high efficiency. Accompanied by the dehydrogenation process, reactive phenyl radicals are passivated by detached H atoms with a 100% yield. The

generated alkynyl radicals via radical transfer subsequently interact with each other, forming alkynyl-Ag-alkynyl bonds. The complex reaction pathways are further corroborated by the convergent STM observations, TPD treatments, TP-XPS measurements and DFT calculations. Finally, we discovered that such inter-molecular radical transfer mechanisms are universal for different on-surface dehydrogenation reactions. Our studies enrich the on-surface synthesis methodologies and develop a pathway for fabricating low-dimensional organic materials.

## Methods

### STM measurements

All the samples were prepared with commercial low-temperature scanning tunneling microscopy (Omicron) with a base pressure better than $1 \times 10^{-10}$ Torr. The single-crystal Ag(111) and Cu(111) substrates (MaTeck, 99.999%) were cleaned by several cycles of Ar$^+$ ions sputtering and subsequent annealing to 750 K prior to deposition of organic molecules. All the images were acquired at 77 K in constant current mode. The STM images were processed using the WSxM software[62]. The Ext-TEB, DBTP, DITP, DAQP, APTP, and BABP molecules were purchased from Innochem Company (purity >95%). The DCTP was provided by Prof. Peinian Liu's group at East China University of Science and Technology (purity >95%). For the TPD/TP-XPS experiments, the Ext-TEB was provided by Svetlana Klyatskaya and Mario Ruben at Karlsruher Institut für Technologie (purity >95%).

### DFT calculations

All DFT calculations were performed with the Vienna Ab-initio Simulation Package (VASP) code[63]. The electron-ion interactions were described by the projector augmented wave (PAW) potentials[64], in which the kinetic cutoff energy for the plane wave basis was set to 400 eV. The exchange-correlation interactions were treated by the Van der Waals density functional (vdWDF) in the version of rev-vdWDF2 developed by Hamada[65]. Such method has been successfully applied in the field of on-surface synthesis, in which the adsorption of hydrocarbons on Ag(111) and Cu(111) can be described accurately[66,67]. The Ag(111) and Cu(111) surfaces were modeled by four layered slabs using a p(8 × 8) supercell. A vacuum layer of 15 Å was employed to avoid periodic image interactions. The Brillouin zone was modeled by the 3 × 3 k-point sampling. Transition states of reactions were identified by using a combination of Climb Image Nudged Elastic Band (CI-NEB) method and Dimer method[68,69]. The CI-NEB method was employed to provide an initial guess of a transition state and the saddle point was further obtained by Dimer calculations. All structures including local minima and saddle points were optimized until the residual force acting on all atoms were below 0.02 eV Å$^{-1}$, except the bottom two layers of metal surfaces which were kept frozen.

### TPD and TP-XPS experiment

All TP-XPS and TPD experiments were performed in a home-built UHV chamber at the Technical University of Munich. In the TPD experiments, the samples were heated with a constant heating rate of 1 K s$^{-1}$ in front of a quadrupole mass spectrometer (QMS) mounted behind a Feulner-cap and the desorbing species were detected upon mass discrimination[70]. The QMS signal was proportional to the rate of desorption. The TP-XPS measurements were carried out using the non-monochromatized Mg Kα line (hν = 1253.6 eV) from a standard X-ray tube and a SPECS Phoibos 100 CCD hemispherical analyzer. All TP-XPS measurements were performed in grazing electron emission geometry. TP-XPS spectra of C 1$s$ and Br 3$d$ were acquired at a constant heating rate of 0.05 K s$^{-1}$, 0.02 K s$^{-1}$, and 0.01 K s$^{-1}$, such that each individual XP spectrum had a temperature resolution of ≈1−4 K. The binding energy of all XP spectra was calibrated against the Ag 3$d_{5/2}$ core level of the Ag(111) substrate at a binding energy of 368.27 eV[71].

## Reporting summary

Further information on research design is available in the Nature Portfolio Reporting Summary linked to this article.

## Data availability

The data that support the findings of this study are available from the corresponding author upon request. Source data are provided with this paper.

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

## Acknowledgements

We acknowledge the financial support by the Fundamental Research Funds for the Central Universities (Grant GK202201001 (Q.L.), GK202203002 (H.L.)), the National Natural Science Foundation of China (22272099 (Q.L.), 22072102 (Q.L.), 51821002 (L.C.), 22202125 (P.H.)), and the Collaborative Innovation Center of Suzhou Nano Science & Technology, the Suzhou Key Laboratory of Surface and Interface Intelligent Matter (Grant SZS2022011 (L.C.)), and the 111 Project. Funding is further acknowledged from the Swedish Research Council and the Göran Gustafsson Foundation for Research in Natural Sciences and Medicine. The computations were enabled by resources provided by the National Academic Infrastructure for Supercomputing in Sweden (NAISS) and the Swedish National Infrastructure for Computing (SNIC), funded by the Swedish Research Council through grant agreements no. 2022-06725 (J.B.) and no. 2018-05973 (J.B.). We thank Svetlana Klyatskaya and Mario Ruben for providing Ext-TEB molecules in TPD/TP-XPS experiments, Pablo Vezzoni and Peter Feulner for support and advise with the TPD/TP-XPS experimental setup and data analysis.

## Author contributions

L.C. and Q.L. designed the entire experiments. J.W., H.Z., and C.D. performed the STM measurements. J.R., J.B., and K.N. performed the DFT calculations. B.Y. and W.Z. conducted the TPD and TP-XPS experiments. D.L., P.L., and C.X. synthesized the precursor molecules. F.A., J.V.B., P.H., and H.L. helped in data analysis. All the authors contribute to the preparation of the manuscript.

## Competing interests

The authors declare no competing interests.
