## [Peer Review File · Nature Communications]

Universal inter-molecular radical transfer reactions on metal surfacesREVIEWER COMMENTS

Reviewer #1 (Remarks to the Author):

The manuscript by Wang and coworkers describes a very interesting effect that has not yet been really highlighted in on-surface synthesis, namely the use of a co-reactant to initiate an intermolecular radical transfer reaction. This study should have a high impact in the field, but also potentially to a much broader chemistry community, because on-surface synthesis often gives access to new and unexpected processes. While on-surface synthesis investigations are often limited to homocoupling reactions, the manuscript demonstrates that the use of two different properly designed reactants, a priori non-interacting, can give rise to intriguing effects.

But in fact the effect of radical transfer in terminal alkyne coupling has been already reported recently (Lawrence, ACSNano 2021, 15, 4937). In addition, this previous work (not yet cited!) strongly suggests that most of the older work on terminal alkyne coupling performed by the authors' group (refs. 47,48) and others (ref. 49) should be properly reassigned because dehydrogenation can probably never occur without radical transfer. Due to the lack of novelty, the manuscript is not acceptable for a publication in Nature Communications. In any case, major revisions need to be first conducted. The experimental work is of high quality and rather straightforward. The theoretical modelling, however, is questionable and certainly too simplistic to encompass the whole complexity of this intriguing phenomena. My criticisms are detailed below:

1) In the introduction, after having detailed the limitations of on-surface chemistry on a metal surface (p.3), it should be noted that they can be partly lifted for decoupled systems or on insulating surfaces, with the demonstration of radical polymerization (Para, Nat. Chem. 2018, 11, 1112) or light-induced reactions (Grossmann, Nat. Chem. 2021, 13, 730), for examples.

2) The two reactants DBTP and Ext-TEB self-assemble into separate domains, and it is said p.6 that the debromination reaction is not affected by the presence of an alkynyl group in vicinity. As a consequence, the debromination should occur in the DBTP-pure domains and therefore lead to Ullmann coupling, which is not the case. So it is clear from the experimental results that both molecules are in close interaction in a mixed-phase during

the reaction. Here a temperature-dependent LEED investigation may be very insightful to observe the phase mixing. But the DFT model of Fig.3 is thus much too simplistic. Also, it is compared p.6 to the calculations of ref.48, which are certainly not valid considering the new report showing that dehydrogenation does not occur in bare terminal alkyne coupling (Lawrence, ACSNano 2021, 15, 4937). In fact, the TPD results of Fig.3 indeed show that there is some dehydrogenation, but the measurement is not quantitative and it can still come from a partial dehydrogenation (at defect, step edges, ...?).

3) The reaction leads to the formation of alkynyl-Ag-alkynyl bonds, but the calculations never include an Ag adatom. The role of the latter may be non-negligible in the reaction process and must be also considered.

4) The paragraph p.9,10 about the “universality” of the process, showing similar results on the amine dehydrogenation reaction, is very interesting, but it is still a bit complex to apprehend, and therefore it requires a much better treatment, i.e. more STM images and more detailed results. I understand the authors’ goal of claiming for universality, but including it in a single SI image is too limited and I suggest to keep this part for a separate manuscript.

5) Investigations including the participation of H atoms are very delicate to handle in UHV and should be considered with care because there may be other sources of hydrogen in the vacuum chamber. In particular, control experiment should be conducted with switching off all ion gauge filaments, see Enderson, ACSNano 2023, 17, 7366. This issue must be properly addressed in the manuscript.

6) Fig. 4: There seems to be an incoherently high number of Br atoms in the STM images, i.e. not only in the DBTP phase with the correct stoichiometry, but also in the Ext-TEB phase. Please comment.

7) Fig. 5: please indicate also here the molecular ratio.

8) Fig. S1e,f: the STM image is fuzzy, some molecules seem to be still mobile, so the model and the intermolecular interactions are hard to understand, please detail.

9) Fig. S13a,b: caption: Cu(111), not Ag(111).

Reviewer #2 (Remarks to the Author):

In the manuscript, the authors report inter-molecular radical transfer reactions with the aid of STM, TPD, XPS and DFT. Upon this, the dehydrogenation of terminal alkynes was promoted with the assistance of radicals from the dehalogenation of halobenzene molecules. In the meanwhile, the dehalogenated halobenzene molecules were passivated by detached hydrogen atoms from alkynes. In addition, such inter-molecular radical transfer mechanism was investigated in other systems, e.g., reactions of amino/imino derivatives, showing its potential universality. Despite its wide exploration in solution-phase chemistry, inter-molecular radical transfer in on-surface synthesis has been rarely reported. Therefore, this work is novel and of great significance, which offers an evidence in real space at atomic precision for the classical inter-molecular radical transfer.

However, some data interpretation and the universality of the intermolecular radical transfer lack strong evidences. More discussions and experimental proofs should be given to further confirm them. A few suggestions are listed below. Once these concerns are satisfactorily addressed, the paper could be recommended for the publication in Nat. Commun.

1. In Fig. S5 and S6, the authors showed “the dehydrogenation of Ext-TEB occurs after annealing the sample at 380 K/400 K”. What are the products in each case? Alkynyl-Ag-Alkynyl linked organometallic motifs or covalent structures by Glaser coupling? The former would indicate that Br adatoms also have catalytic performance on the dehydrogenation of terminal alkyne.
2. The author highlighted a lot of times that “Debrominated phenyl radicals are passivated by H atoms with 100% reaction selectivity”, which is indeed the key point of the reaction mechanism of inter-molecular radical transfer. However, the conventional STM images do not have enough resolution to identify the H saturation. Nc-AFM with a CO-functionalized probe will help.
3. According to the inter-molecular radical transfer mechanism proposed by the authors, the terminal alkyne should be very close to the dibrominated halobenzene to make the reaction occur. This implies that if the molecular coverage is extremely low or extremely high the reaction yield should be lower because the chance that two reactants meet together is reduced. Did the authors perform these experiments?
4. As for the universality, the reviewer suggests the authors also use a small molecule like $\text{CH}_3\text{CH}_2\text{Br}$ as the radical source, which has a sufficient mobility on the surfaces. According

to the mechanism proposed by the authors, it could also promote the dehydrogenation of terminal alkynes. On the contrary, if a halobenzene molecule, as the radical source, is too large, it may not promote the dehydrogenation of terminal alkynes because of the limited diffusion on surfaces. The author should consider to accomplish these control experiments to strength the conclusion.

5. The bibliography should be expanded, because plenty of similar studies on radical promoted dehydrogenation of terminal alkynes have been reported, though without direct evidences, e.g. ACS Nano 2021, 15, 3, 4937–4946; Phys. Chem. Chem. Phys. 2018, 20, 11081–11088. Angew. Chem. Int. Ed. 2017, 56, 4762– 4766. These references could be in turn used to support the conclusion of this work.

Minor comments:

1. Please check the English grammar in the manuscript carefully.
2. Fig. 2d. some bright dots filling in the self-assembled island should be Ag adatoms, as seen in ACS Nano 2018, 12, 2267–2274; Chem. Eur. J. 2017, 23, 6190-6197. Therefore, the author should be more careful measuring the length of a monomer.
3. “significant desorption is observed for both the alkynyl-Ag species and passivated phenylene”. Is it possible to desorb a alkynyl-Ag species from a Ag(111) surface?
4. Page 3. “(2) Intermolecular radical transfer should be favored energetically, with the energy barrier of the radical transfer being lower than or comparable to that of the radical generation.” This would only determine the radical transfer is not the rate-limiting step, instead of implying an intermolecular radical transfer cannot happen.

Reviewer #3 (Remarks to the Author):

The field of on-surface-synthesis has been highly active within experimental surface science for the past 1.5 decades, using a combination of molecular resolution imaging techniques, chemical spectroscopy and theoretical modelling to address how organic molecules adsorbed on solid surfaces may react to form covalently bonded organized structures.

Here, the authors add significant new insights into possible principles of on-surface-synthesis by proposing a new radical transfer mechanism to facilitate dehydrogenation and molecular coupling. Their experiments involve co-adsorption of a three-spoke terminal

alkyne with a tri-phenyl halide on a surface of silver. Upon heating the substrate, they find the co-adsorbed three-spoke alkynes to react differently than when adsorbed alone, and form an extended network attributed to dehydrogenated three-spoke alkynes interconnected by coordinating to Ag atoms (as observed by molecular level scanning tunneling microscopy). The authors attribute this unexpected finding to a radical transfer mechanism where dissociation of the halogen (Br) atom from the tri-phenyl halide, creates a reactive tri-phenyl radical which abstracts hydrogen from the terminal alkyne, leading to the coordination complex. This proposed novel reaction mechanism is corroborated by DFT calculations of the reaction path as well as an impressive array of control and support experiments which all form a consistent picture as far as this referee can judge. The paper builds on a large amount of previous knowledge for the on-surface reaction mechanisms of alkynes (Glaser coupling) and phenyl halides (Ullmann coupling) which makes it somewhat challenging to read, but the authors do a good job of referring the existing literature to make the story tractable.

The evidence for the creation of the intermediate phenyl radical and the eventual claimed saturation of this radical by the hydrogen abstracted from the terminal alkyne is overall circumstantial, but well supported by the consistent picture leading to plausible explanations for all the observed phenomena, e.g for the observed threshold temperatures for radical transfer reactions (or lack thereof) for I, Br and Cl based tri-phenyl halides with different activation energies for reaction. My only question concerns the control experiments reported in Fig. 4 using a stoichiometric excess of the three-spoke alkyne: What happens if the reaction is instead performed with an excess of the tri-phenyls? Will one in this situation observe molecules created by phe-phe coupling since there is then not enough hydrogen abstracted from the available alkynes to passivate all tri-phenyl radicals? Discussion of such data would provide further support for the proposed reaction mechanism.

Overall, I find the paper should be of interest to organic chemists in general as it reports new insights into radical transfer reactions for molecules adsorbed on metal surfaces (where the interaction of the radical with the surface electronic structure has been thought to suppress radical transfer reactions). The paper is particularly interesting for readers in the

field of molecular nanoscience, since the presented data for other (amine-containing) compounds suggest the radical transfer mechanism induced by co-adsorption with a phenyl-halide might be a quite general avenue to facilitate hydrogen abstraction, opening up for new pathways within on-surface-synthesis.

Reviewers' comments:

Reviewer #1 (Remarks to the Author):

Comments:

(I) The manuscript by Wang and coworkers describes a very interesting effect that has not yet been really highlighted in on-surface synthesis, namely the use of a co-reactant to initiate an intermolecular radical transfer reaction. This study should have a high impact in the field, but also potentially to a much broad chemistry community, because on-surface synthesis often gives access to new and unexpected processes. While on-surface synthesis investigations are often limited to homocoupling reactions, the manuscript demonstrates that the use of two different properly designed reactants, a priori non-interacting, can give rise to intriguing effects.

Author reply: We thank the reviewer for the careful reviewing, and the appropriate extracting of our research highlights.

(II) But in fact the effect of radical transfer in terminal alkyne coupling has been already reported recently (Lawrence, ACS Nano 2021, 15, 4937). In addition, this previous work (not yet cited!) strongly suggests that most of the older work on terminal alkyne coupling performed by the authors' group (refs. 47,48) and others (ref. 49) should be properly reassigned because dehydrogenation can probably never occur without radical transfer.

Author reply: We thank the reviewer for pointing out the missing citation to the important reference (ACS Nano **2021**, 15, 4937). In that paper, Lawrence *et al.* employed bond-resolving STM imaging, revealing a non-dehydrogenative head-to-head alkyne coupling scenario on noble metal surfaces, which differs from the dehydrogenative homo-coupling of alkynes reported by some of the authors involved in our study (*Nat. Commun.* **2012**, 3, 1286; *J. Phys. Chem. C* **2014**, 118, 3181) and by other research groups (*J. Phys. Chem. C* **2013**, 117, 18595). It should, however, be noted that in the study by Lawrence *et al.* from 2021 the reaction was investigated on Au(111) and not Ag(111). Recent study by Chi Zhang *et al.* reported that the coupling of alkynes on Ag(111) leads to the formation of enyne and cumulene products investigated by tip-enhanced Raman spectroscopy measurements (*J. Am. Chem. Soc.* **2021**, 143,

9461). The contradictory conclusions and the raised comments prompted the involved researchers (some of whom are involved in the present work) to reexamine the same system (Ext-TEB/Ag(111)) with high-resolution non-contact atomic force microscopy (nc-AFM). Latest results with increased data precision, notably regarding the carbon-carbon bonding motifs (*unpublished*) indicate indeed that non-dehydrogenative coupling reactions of terminal alkynes dominate on Ag(111). Moreover, additional DFT modeling (*unpublished*) performed by one of the authors involved in this work also demonstrate that the non-dehydrogenative coupling of terminal alkynes are the most favorable pathway. The calculations furthermore signal that the reaction is initiated with C-C coupling, as previously concluded (*J. Phys. Chem. C* **2014**, 118, 3181), and then followed by hydrogen transfer steps. The entire reinvestigation of the surface-confined terminal alkyne coupling reaction by the combination of STM/nc-AFM investigations and DFT calculations are currently written up and will be submitted as an independent manuscript shortly.

The scope of this study is the radical-induced dehydrogenation of terminal alkynes and we would like to focus on this aspect rather than a re-interpretation of the nature of the coupling motifs in the absence of radical transfer. Nevertheless, we have nuanced our presentation of the results, in which the scenario where the terminal alkynes react without the presence of a radical being referred to simply as a “non-dehydrogenative coupling between terminal alkynes”. Moreover, the products formed via the coupling of terminal alkynes without the presence of a radical shown in the scheme is redrawn to an enyne product.

Overall, we agree with the referee that the description of the previously reported coupling reaction of terminal alkynes should be adequately reconsidered. We would like to emphasize that the reassessment of the alkyne coupling does not influence our findings and conclusions regarding the radical transfer reaction on surfaces. Actually, the reassessment makes our research even more intriguing, as the radical transfer mechanism alters the reaction pathway from producing enyne products via non-dehydrogenative homo-coupling to forming alkynyl-Ag-alkynyl products initiated by dehydrogenation of alkyne groups.

To avoid any potential misunderstandings, we made the following changes in the revised manuscript and Supporting Materials.

Added ref. 23, 24, 32 and 46 in the revised manuscript.

Added ref. 1-4 in the revised supporting materials.

Modified corresponding structural models shown in Fig. 1, Fig. 5, Fig. 6, Supplementary Fig. 1 and Graphical abstract.

Page 2, Line 9, modification “With the assistance of aryl halide precursors, the coupling of terminal alkynes can be steered from non-dehydrogenated to dehydrogenated products, resulting in alkynyl-Ag-alkynyl bonds.”

Page 3, Line 26, modification “The formed radicals facilitate dehydrogenation reactions of alkyne groups, and alter reaction pathways of alkyne groups from non-dehydrogenated to dehydrogenated coupling reactions. Specifically, temperatures required to initiate the coupling of alkyne groups reduce from 400 K (without aryl halide precursors) to 340 K when taking aryl bromide as the radical initiator, and to below room-temperature (RT) when taking aryl iodides as the radical initiator on Ag(111) (Fig. 1).”

Page 4, Line 5, modification “In contrast, without the assistance of aryl halide precursors, alkyne derivatives engage in non-dehydrogenative coupling reactions on Ag(111) (Fig. 1), as previously shown. ^{23,24,32,46}”

Page 4, Line 19, modification “Coupling of Ext-TEB molecules take place after annealing at 400 K (Supplementary Fig. 1c, d). ^{23,24,32,46}”

Page 4, Line 23, modification “The temperatures needed to trigger coupling of Ext-TEB and debromination of DBTP are in excellent agreement with previous reports. ^{24,47}”

Page 5, Line 5, modification “which strikingly differs from the coupling reactions of isolated alkynes on silver (Supplementary Fig. 1d). ^{23,24,32,46}”

Page 6, Line 2, modification “The temperature needed to trigger the coupling of Ext-TEB is reduced markedly from 400 to 340 K.”

Page 7, Line 23, delete “In contrast, two isolated alkynyl groups couple with each other firstly, and then undergoes two subsequent dehydrogenation processes on Ag(111), resulting in the formation of alkynyl-alkynyl bonds.”

Page 7, Line 29, modification “the temperature needed to trigger the coupling reaction of Ext-TEB decreased markedly;”

Page 8, Line 17, modification “Remaining terminal alkynes maintain pristine (Fig. 4c), as the temperature required to trigger the coupling of Ext-TEB without the assistance of an aryl halide

precursor is much higher (Supplementary Fig. 1d).”

Page 8, Line 24, modification “The control experiment not only clarifies that the presence of phenyl radicals reduces the temperature required to initiate the coupling reaction of alkynes,”

Page 9, Line 12, modification “coupling of Ext-TEB takes place after annealing at 400 K, and the alkynyl-Ag-alkynyl products are not observed (Fig. 5g).”

Page 9, Line 28, modification “On the other hand, the reaction temperature for dechlorination of DCTP is significantly higher than that for coupling of isolated Ext-TEB, which inhibits intermolecular radical transfer reactions. Consequently, the formation of alkynyl-Ag-alkynyl products is prohibited (Fig. 5g and Fig. 6f).^{23,24,32,46}”

Supplementary Information, Page S2, Line 6, modification “The coupling of Ext-TEB takes place after annealing the sample at 400 K (Supplementary Fig. 1c).¹⁻⁴”

Supplementary Information, Page S10, Line 12, modification “In contrast, Ullmann reactions and coupling reaction of alkynes take place in DBTP/Ag(111) and Ext-TEB/Ag(111) systems, respectively. The formed products hardly desorb from the surface.”.

(III) Due to the lack of novelty, the manuscript is not acceptable for a publication in Nature Communications. In any case, major revisions need to be first conducted. The experimental work is of high quality and rather straightforward. The theoretical modelling, however, is questionable and certainly too simplistic to encompass the whole complexity of this intriguing phenomena.

Author reply: The concept of radical chemistry has been widely applied in solution. On surfaces, radical-induced reactions are rarely reported. Recently, Lawrence *et al.* (*ACS Nano* **2021**, *15*, 4937) demonstrated that coupling of two terminal alkynes can be attributed to one of the two possible mechanisms: (1) radical transfer via hydrogen migrations; (2) coadsorbed atomic bromine catalysis (also proposed by Liu *et al.*, *Phys. Chem. Chem. Phys.* **2018**, *20*, 11081). Meanwhile, other groups proposed alternative candidate mechanisms, including hybrid Ag species catalysts (*Nat. Chem.* **2018**, *10*, 296) and the influence of molecular configurations (*Surf. Sci.* **2023**, 727, 122180). To summarize, the mechanisms of dehydrogenative coupling reaction of terminal alkynes with the presence of halogen-functionalized reactants has been long-standing debated. However, the radical transfer mechanism has not been confirmed

experimentally.

In this article, we provide strong and systematic evidence to elucidate the radical transfer mechanism using two different functional reactants. The radical transfer is evidenced by the passivation of the phenyl radicals via hydrogen transfer, and the dehydrogenative reaction pathways of coupling reaction of alkynes. Moreover, we point out the universality of such mechanisms. We therefore believe that our systematic work, combining of various surface techniques (STM, TPD, TP-XPS) and DFT calculations, will be of general interest for surface scientists and chemists.

Regarding the calculations, we have carefully read the reviewer's comments and conducted additional DFT calculations involving Ag adatoms to increase the complexity of the model. Please see the reply of question 3 of referee 1 for details. We would, however, like to emphasize that the theoretical work presented in our original submission addresses how the halide compounds assist the dehydrogenation; the dehydrogenation is driven by the radical sites resulting from dehalogenation rather than by the abstracted halogens.

In the following, we address all the other questions:

My criticisms are detailed below:

1, In the introduction, after having detailed the limitations of on-surface chemistry on a metal surface (p.3), it should be noted that they can be partly lifted for decoupled systems or on insulating surfaces, with the demonstration of radical polymerization (Para, Nat. Chem. 2018, 11, 1112) or light-induced reactions (Grossmann, Nat. Chem. 2021, 13, 730), for examples.

Author reply: We thank the reviewer for the comment. As suggested by the referee, we added a sentence in the introduction:

Added **ref. 44 and 45** in the revised manuscript.

Page 3, Line 21, modification “**Such spurious effects can be partly voided by performing chemical reactions on non-metal surfaces.**^{44,45}”

2a, The two reactants DBTP and Ext-TEB self-assemble into separate domains, and it is said p.6 that the debromination reaction is not affected by the presence of an alkynyl group in vicinity. As a consequence, the debromination should occur in the DBTP-pure domains and

therefore lead to Ullmann coupling, which is not the case. So it is clear from the experimental results that both molecules are in close interaction in a mixed-phase during the reaction. Here a temperature-dependent LEED investigation may be very insightful to observe the phase mixing.

Author reply: We agree with the referee that the diffusion abilities of the precursors are important, as it determines whether the initial states of the DFT calculation shown in Fig. 3 can be achieved. Actually, separate DBTP and Ext-TEB domains are observed at 77 K with high coverage (Fig. 2a). On the other hand, the radical transfer reaction shown in Fig. 2b takes place at 340 K, at which the precursor molecules indeed diffuse sufficiently on Ag(111). Support for this is provided in the following:

1. As shown in Reply Fig. 1a and b, fuzzy patterns are observed after depositing submonolayer Ext-TEB (0.2 ML) and DBTP (0.2 ML) separately on Ag(111). The self-assemble islands are stabilized after the coverage increases to about 1 ML (Supplementary Fig. 1). Moreover, after co-depositing 0.2 ML Ext-TEB and 0.2 ML DBTP molecules on Ag(111), fuzzy patterns are obtained at 77 K, as shown in Reply Fig. 1c. Increasing the total coverage close to 1 ML leads to the stabilization of the Ext-TEB and DBTP domains (Reply Fig. 1d). Despite of that, one can still observe fuzzy areas between adjacent domains. The fuzzy patterns shown in Reply Fig. 1 suggest that at 77 K only weak molecule (Ext-TEB/DBTP)-substrate and molecule (Ext-TEB/DBTP)-molecule (Ext-TEB/DBTP) interactions are exerted.

2. A previous study has demonstrated that depositing aryl bromide precursors on Ag(111) leads to the coexistence of self-assemble islands and fuzzy regions at 97 K on Ag(111) (*ACS Nano* **2018**, *12*, 2267). The size of fuzzy regions increases significantly after annealing the sample to 113 K, due to the weak halogen bonding (*Nat. Commun.* **2020**, *11*, 5630; *J. Phys. Chem. C* **2013**, *117*, 302). Meanwhile, given the weak C-H $\cdots\pi$ interactions between terminal alkynes (*ACS Nano* **2012**, *6*, 1, 566; *Phys. Chem. Chem. Phys.* **2011**, *13*, 13873), the decomposition of Ext-TEB assembly islands at elevated temperatures is foreseeable. Moreover, phenylacetylene molecules diffuse sufficiently already below 120 K on Cu(100) (*Nat. Commun.* **2018**, *9*, 3113; *Sci. Rep.* **2013**, *3*, 2102).

It is therefore safe to conclude that both Ext-TEB and DBTP precursor molecules diffuse sufficiently at the reaction temperature (340 K), such that the initial state shown in Fig. 3 can

easily be populated.

Reply Fig. 1 | The sufficient diffusion of Ext-TEB and DBTP molecules on Ag(111). **a** and **b**, STM topographic images after separately depositing DBTP (0.2 ML) and Ext-TEB (0.2 ML) on Ag(111) held at RT, respectively. **c**, STM topographic images after the co-deposition of 0.2 ML Ext-TEB and 0.2 ML DBTP on Ag(111) held at RT. **d**, STM topographic images after the co-deposition of 0.4 ML Ext-TEB and 0.6 ML DBTP on Ag(111) held at RT. Tunneling parameters are $I_t = 100$ pA and $V_b = -1$ V for all the STM images. Image sizes are 30×30 nm² for all the STM images. C, Br and H atoms are represented by gray, brown and white circles, respectively.

To avoid confusion, we added corresponding descriptions:

Added ref. 5, 9 and 17-21 in the revised supporting materials.

Page 6, Line 20, added “As both precursors diffuse sufficiently at 340 K on Ag(111) (see Supplementary Fig. 11 for details)”.

Supplementary Information, Page S15, Line 1, added “**11. The diffusion of Ext-TEB and DBTP on Ag(111)**”

The diffusion abilities of precursors are important, as it determines whether the initial

states of the DFT calculation shown in Fig. 3 can be achieved. When the radical transfer reaction occurs (340 K), the precursor molecules indeed diffuse sufficiently on Ag(111). Support for this is provided in the following:

1. As shown in Supplementary Fig. 11a and b, fuzzy patterns are observed after depositing submonolayer Ext-TEB (0.2 ML) and DBTP (0.2 ML) separately on Ag(111). The self-assemble islands are stabilized after the coverage increases to about 1 ML (Supplementary Fig. 1). Moreover, after co-depositing 0.2 ML Ext-TEB and 0.2 ML DBTP molecules on Ag(111), fuzzy patterns are obtained at 77 K, as shown in Supplementary Fig. 11c. Increasing the total coverage close to 1 ML leads to the stabilization of the Ext-TEB and DBTP domains (Supplementary Fig. 11d). Despite of that, one can still observe fuzzy areas between adjacent domains. The fuzzy patterns shown in Supplementary Fig. 11 suggest that at 77 K only weak molecule (Ext-TEB/DBTP)-substrate and molecule (Ext-TEB/DBTP)-molecule (Ext-TEB/DBTP) interactions are exerted.

2. A previous study has demonstrated that depositing aryl bromide precursors on Ag(111) leads to the coexistence of self-assemble islands and fuzzy regions at 97 K on Ag(111).⁹ The size of fuzzy regions increases significantly after annealing the sample to 113 K, due to the weak halogen bonding.^{5,17} Meanwhile, given the weak C-H $\cdots\pi$ interactions between terminal alkynes,^{18,19} the decomposition of Ext-TEB assembly islands at elevated temperatures is foreseeable. Moreover, phenylacetylene molecules diffuse sufficiently below 120 K on Cu(100).^{20,21}

It is therefore safe to conclude that both Ext-TEB and DBTP precursor molecules diffuse sufficiently at the reaction temperature (340 K), such that the initial state shown in Fig. 3 can easily be populated.”

Added Supplementary Fig. 11.

We did not perform TP-LEED experiments as we believe the diffusion of precursor molecules at the reaction temperature can be confirmed by Reply Fig. 1.

2b. But the DFT model of Fig.3 is thus much too simplistic.

Author reply: We thank the reviewer for the comments. Fig. 3 presents the most energetically

avored pathway for achieving dehydrogenated terminal alkyne and H-saturated phenyl group via radical transfer, agreeing well with experimental observations. To enhance the complexity of the applied model, supplementary calculations and descriptions are performed to elucidate the reaction pathway involving Ag adatoms. The details can be seen in the reply of question 3 of referee 1 below.

2c. Also, it is compared p.6 to the calculations of ref.48, which are certainly not valid considering the new report showing that dehydrogenation does not occur in bare terminal alkyne coupling (Lawrence, ACS Nano 2021, 15, 4937). In fact, the TPD results of Fig.3 indeed show that there is some dehydrogenation, but the measurement is not quantitative and it can still come from a partial dehydrogenation (at defect, step edges, ...?).

Author reply: We agree with the referee that the homocoupling of terminal alkynes without the presence of halogenated precursors leads to the formation of enyne products (See the reply of question (II) of referee 1 for details). Latest DFT calculations by one of the authors (they will be submitted for publication, combined with nc-AFM results in a separate manuscript) and previous reports (*Nat. Chem.* **2016**, *8*, 678; *J. Am. Chem. Soc.* **2021**, *143*, 9461; *J. Am. Chem. Soc.* **2015**, *137*, 1833; *ACS Nano* **2021**, *15*, 4937) reveal that the non-dehydrogenative pathway also undergoes C-C couplings first, similar to the dehydrogenative reaction pathway (*J. Phys. Chem. C* **2014**, *118*, 3181). After that, intra-molecular hydrogen transfers take place in the non-dehydrogenative reaction pathway, which differs slightly from the previously reported hydrogen detachments in the dehydrogenative reaction pathway (*Nat. Commun.* **2012**, *3*, 1286).

In our bi-component system, DFT calculations reveal that the reaction barrier is determined by the competition of the detachment of halogen atom and the hydrogen transfer. The scission of hydrogen atoms from alkynyl groups with the affinity of phenyl radicals exhibits an activation barrier of 0.78 eV. For reaction without the assistance of aryl halide precursors, the C-C coupling (*Nat. Chem.* **2016**, *8*, 678; *J. Am. Chem. Soc.* **2015**, *137*, 1833; *J. Phys. Chem. C* **2014**, *118*, 3181) is energetically more favorable compared to direct C-H activations of a terminal alkyne group on Ag(111) (*J. Am. Chem. Soc.* **2022**, *144*, 10282; *J. Phys. Chem. C* **2014**, *118*, 3181). The energy barrier of the radical transfer process is, however, lower than that for the C-C coupling of two terminal alkynes, leading to the formation of alkynyl-Ag-alkynyl

motifs.

Moreover, we agree with the reviewer's comments about the TPD results.

To avoid confusion, we made the following corrections in the revised version:

Added ref. 32, 48 and 59 in the revised manuscript.

Page 7, Line 3, modification “**For reaction without the assistance of aryl halide precursors, the C-C coupling^{32,53,59} is energetically more favorable compared to the direct C-H activation of a terminal alkyne group on Ag(111)^{48,53}. The energy barrier of the radical transfer process is, however, lower than that for the C-C coupling of two terminal alkynes.**”

Page 8, Line 8, modification “**This phenomenon probably arises from that certain side reactions are suppressed by the radical transfer reactions.**”

3, The reaction leads to the formation of alkynyl-Ag-alkynyl bonds, but the calculations never include an Ag adatom. The role of the latter may be non-negligible in the reaction process and must be also considered.

Author reply: We reconsidered the radical transfer process (scenario 1 and 2) and the subsequent formation of alkynyl-Ag-alkynyl products after the radicals transfer from phenyl to alkyne (scenario 3) by involving Ag adatoms.

In scenario 1, we considered isolated Ag adatoms serving as the active centre for the direct C-H bond dissociation of an isolated terminal alkyne. Actually, the previous report points out that the Ag adatom exhibits limited activity towards dehydrogenation of alkynes (*J. Phys. Chem. C* **2014**, *118*, 3181). The formation of alkynyl-Ag-alkynyl products therefore cannot be ascribed to Ag adatoms catalysts.

In scenario 2, we considered that Ag adatom participates in the radical transfer process. By affinity to phenyl radicals, Ag adatoms exist as hybrid Ag in Ph-Ag species. As shown in Supplementary Fig. 10, the overall energy barrier for the C-H activation of terminal alkyne catalysed by the Ag hybrid species is 1.29 eV, therefore less energetically favourable than the proposed radical transfer mechanism in the absence of adatoms (0.78 eV).

In scenario 3, supplementary calculations were performed to elucidate the reaction pathway of the formation of alkynyl-Ag-alkynyl products. As shown in Reply Fig. 2, two isolated alkynyl radicals are incorporated with a silver adatom to form the alkynyl-Ag-alkynyl

linkage by overcoming an activation energy of 0.25 eV, implying that the alkynyl-Ag-alkynyl motif can readily be formed as soon as the radical transfer process is completed. The low barrier of the formation of alkynyl-Ag-alkynyl motif explains why alkynyl radicals are not observed on Ag(111) experimentally.

In summary, Ag adatoms exhibit limited catalytic activity towards the dehydrogenation of terminal alkynes, while the formation of organometallic structures can be easily achieved under the presence of Ag adatoms on Ag(111).

Reply Fig. 2. | Formation of alkynyl-Ag-alkynyl bridging motifs on Ag(111). **a**, The reaction pathway for the formation of alkynyl-Ag-alkynyl linkage on Ag(111). **b**, the corresponding energy profiles. Ag, Ag adatom, C and H atoms are represented by silver, blue, gray and white circles, respectively.

We have made the corresponding additions:

Page 6, Line 8, added “An earlier study already pointed out that the Ag adatom exhibits limited activity regarding the dehydrogenation of alkynes.⁵³”

Page 7, Line 21, added “Two alkynyl radicals are subsequently linked by a silver adatom, forming the alkynyl-Ag-alkynyl motif (step III in Fig. 3c, the detailed process is provided in Supplementary Fig. 13), which differs from the coupling reaction of two isolated alkyne groups.”

Supplementary Information, Page S18, Line 1, added “**12. The reaction pathway to form the alkynyl-Ag-alkynyl bridging motifs**

The formation of alkynyl-Ag-alkynyl motif after the radical transfer process was studied by DFT calculations. As shown in Supplementary Fig. 13a, two isolated alkynyl radicals are incorporated with a silver adatom to form the alkynyl-Ag-alkynyl linkage by overcoming an activation energy of 0.25 eV (Supplementary Fig. 13b). Such energy barrier is much lower than that of debromination of bromobenzene (1.05 eV) and the radical transfer process (0.78 eV). Consequently, the alkynyl-Ag-alkynyl organometallic structure can readily be formed as soon as the radical transfer completes. The low barrier of the formation of alkynyl-Ag-alkynyl motif explains why alkynyl radicals are not observed on Ag(111) experimentally.”

Added Supplementary Fig. 13 (also see Reply Fig. 2).

4, The paragraph p.9,10 about the “universality” of the process, showing similar results on the amine dehydrogenation reaction, is very interesting, but it is still a bit complex to apprehend, and therefore it requires a much better treatment, i.e. more STM images and more detailed results. I understand the authors’ goal of claiming for universality, but including it in a single SI image is too limited and I suggest to keep this part for a separate manuscript.

Author reply: Thanks for the constructive suggestions. We thus describe control experiments with one of the precursors in details (Reply Fig. 3, which is same to Supplementary Fig. 20), and only report the phenomenon of the control experiments with other precursors in Tab. 1 in the manuscript:

Added ref. 23-25 in the revised supporting materials.

Page 10, Line 19, added “For example, depositing 4-amino-p-terphenyl (AFTP) on Cu(111) leads to fuzzy STM patterns, suggesting pristine AFTP monomers diffuse quickly on Cu(111) at 77 K. The reaction of AFTP takes place upon annealing at 370 K. In contrast, depositing 4'-Bromo-1,1'-biphenyl-4-amine (BBPA) on Cu(111) at RT stimulates dehydrogenation reactions of amino groups, accompanied by the hydrogen passivation of debrominated phenyl radicals. The details of the control experiment are clarified in Supplementary Fig. 20.”

Supplementary Information, Page S28, Line 2, added “In order to study the generality of the inter-molecular radical transfer mechanism, control experiments are conducted.

Firstly, we choose 4-amino-p-terphenyl (AFTP) as precursor molecule (The structural model is shown in the inset of Supplementary Fig. 20a). After depositing AFTP on a Cu(111) surface held at RT, fuzzy patterns are observed, indicating the fast movement of adsorbed AFTP (Supplementary Fig. 20a). This phenomenon aligns with previous reports that the amino groups remain intact on Cu(111) at RT.²³ Dimer products form after annealing the sample at 370 K for 10 minutes, as seen in Supplementary Fig. 18b, which can be ascribed to the formation of N-Cu-N bonds.²³⁻²⁵ The structural model is shown in the inset of Supplementary Fig. 20b.

In order to study the influence of aryl bromides on the reaction of amino derivatives, 4'-Bromo-1,1'-biphenyl-4-amine (BBPA) is chosen as precursor molecule (The structural model is shown in the inset of Supplementary Fig. 20c). Supplementary Fig. 20c shows a representative STM image after depositing BBPA onto Cu(111) kept at RT. Surprisingly, BBPA dimers are observed, indicating the formation of N-Cu-N bonds. Br atoms are observed in-between the dimers, suggesting the detachment of Br atoms from BBPA. Importantly, the Ullmann coupling reactions are prevented, suggesting that the resulting radical sites of BBPA are passivated by hydrogen atoms. The corresponding structural model of the dimer products is shown in the inset of Supplementary Fig. 20d. Both the occurrence of dehydrogenative reaction at RT and the passivation of radical sites suggest the occurrence of the radical transfer reaction.”

Added Supplementary Fig. 20 (also see Reply Fig. 3).

Reply Fig. 3 | Reaction behavior of APTP and BBPA on Cu (111). **a**, STM topographic image after depositing APTP on Cu(111) held at RT. The inset gives the structural model of APTP. **b**, STM topographic image after annealing the surface shown in **a** at 370 K for 10 minutes. Highly resolved STM image and corresponding structural model are shown in the inset. **c**, STM image after depositing BBPA on Cu(111) held at RT. Structural model of BBPA is given in the inset. **d**, High-resolution STM image of **c**. The inset shows the structural model of the dimer product. Tunneling parameters are $I_t = 100$ pA and $V_b = -1$ V for all the STM images. Image sizes are 100×100 nm² for **a**; 30×30 nm² for **b** and **c**; and 30×30 nm² for **d**. Br, C, Cu, N and H atoms are represented by brown, gray, yellow, blue and white circles, respectively.

5, Investigations including the participation of H atoms are very delicate to handle in UHV and should be considered with care because there may be other sources of hydrogen in the vacuum chamber. In particular, control experiment should be conducted with switching off all ion gauge filaments, see Anderson, ACS Nano 2023, 17, 7366. This issue must be properly addressed in the manuscript.

Author reply: We agree with the referee that it is important to trace the H source that passivated phenyl radicals. We believe that the influence of the residue H source in the vacuum chamber is negligible for the reaction, because:

Firstly, no obvious hydrogen gas signals are observed on bare Ag(111) in TPD signals, as shown in Fig. 3d in the main manuscript. However, we cannot make definite conclusions solely based on TPD results due to the relative low sensitivity.

Secondly, as suggested by the referee, we performed the same reaction with the gauge switched off. Similar results were observed, suggesting passivated H atoms mainly come from detached hydrogen atoms from alkyne groups.

Thirdly, to conclusively rule out the influence of the residual H source in the vacuum chamber, control experiments are performed. We deposited 2-bromo-1,3,5-triphenylbenzene (BTB, the structural model is shown in the inset of Reply Fig. 4a) on Ag(111) at RT and followed by annealing to 370 K for 30 min with the ion gauge switched on. As shown in Reply Fig. 4a, two kinds of triangle-shaped monomers are observed. In type I monomer (Reply Fig. 4b), one of the benzene rings is much brighter than the other two, indicating the generation of phenyl radicals of BTB. The formed radicals cannot undergo subsequent Ullmann reaction due to the steric hindrance. The interaction between phenyl radicals and the substrate leads to the different apparent height of the benzene rings of the monomer. In type II monomer, the monomers exhibit planar configurations (Reply Fig. 4c). This is because radicals of type II monomers are passivated by H atoms in the vacuum chamber. It is noteworthy that once the radicals are passivated, the monomers tend to form self-assembly islands (blue shadow in Reply Fig. 4a), as the interaction between the passivated monomers and the substrate decreases.

To investigate the passivation of phenyl radicals by H atoms in the vacuum chamber, systematical annealing experiments are performed, as shown in Reply Fig. 5. Only phenyl radicals are observed by initial adsorption of BTB on Ag(111) held at RT. The same phenomenon is observed by annealing the sample at 340 K and 370 K for 10 mins (with both gauge on and off). After annealing the surface at 370 K for 30 mins, few islands are observed (one island is observed in several large-scale STM images), suggesting that only a small number of radicals are passivated by H atoms. The quantitative analysis of the passivation is shown in Reply Tab. 1. After annealing the sample at 370 K for 30 min, only 0.15% radicals are passivated with gauge on, and 0.1% radicals are passivated with gauge off. In contrast, in the bi-component system shown in Fig. 2, as soon as phenyl radicals are generated (at 340 K), they are passivated by hydrogen atoms through the radical transfer mechanism with a yield of

100%.

It is therefore safe to conclude that the passivation of radicals by H atoms in the vacuum chamber is negligible at the reaction temperature (340 K).

Reply Fig. 4 | Reaction behavior of BTB on Ag(111). **a**, Large-scale STM image after depositing BTB on Ag(111) held at RT and followed by annealing at 340 K for 30 min. The structural model of BTB is given in inset. **b** and **c**, High-resolution STM images of the region marked as red and blue rectangles in **a**, corresponding molecular models are shown in right panels, respectively. Tunneling parameters are $I_t = 100$ pA and $V_b = -1$ V for all the STM images. Image sizes are 20×20 nm² for **a**; and 2.5×2.5 nm² for **b** and **c**. Br, C and H atoms are represented by brown, gray and white circles, respectively.

Reply Fig. 5 | Gradual passivation of BPB. **a-d**, STM images of BPB on the Ag(111) surfaces under various environmental conditions with the ion gauge filaments turned on. **e-h**, STM images of BPB on the Ag(111) surfaces under various environmental conditions with the ion gauge filaments turned off. Tunneling parameters are $I_t = 100$ pA and $V_b = -1$ V for all the STM images. Image sizes are 100×100 nm² for all images.

Time (min)		0	10	30	60
340 K	On (%)	0.00	0.00	0.00	
	Off (%)	0.00	0.00	0.00	
370 K	On (%)	0.00	0.01	0.10	0.15
	Off (%)	0.00	0.00	0.08	0.11

Reply Tab. 1 Yield of radical passivation. On/off means ion gauge filaments turned on and off.

We added one sentence in the revised manuscript:

Page 5, Line 22, added “It is also noteworthy that similar phenomena are observed by performing the same experimental procedure with the ion gauge switched off, suggesting that the residue of H atoms in the chamber does not influence the reaction.”

As the data shown in Reply Fig. 4 and 5 and Reply Tab. 1 will be submitted in another separate manuscript, we did not add these data in the revised manuscript.

6, Fig. 4: There seems to be an incoherently high number of Br atoms in the STM images, i.e. not only in the DBTP phase with the correct stoichiometry, but also in the Ext-TEB phase. Please comment.

Author reply: The observation of the excess of Br adatoms is rather common in Ullmann reactions (*J. Phys. Chem. C* **2020**, *124*, 16415; *J. Phys. Chem. C* **2021**, *125*, 11454; *ACS Nano* **2019**, *13*, 9270). However, previous papers did not provide definitive explanations for such phenomenon.

In our case, the desorption of molecules in the bi-component system at the reaction

temperature is not obvious according to our TP-XPS spectra. Moreover, the adsorption of Br atoms generated in the organic evaporator can be ruled out, as no Br adatoms are detected by solely depositing DBTP on Ag(111) held at RT (see Supplementary Fig. 1e). We therefore believe such phenomenon may arise from one or several of the following reasons: (1) there may exist the self-assembly islands of passivated DBTP without the decoration of Br adatoms, which we did not observe. (2) Plenty of monomers adsorb along the step edges. (3) Small amount of passivated bromobenzene molecules desorb from surface, and the desorption amount is lower than the detection limit of TP-XPS.

7, Fig. 5: please indicate also here the molecular ratio.

Author reply: In Fig. 5a and b, the stoichiometric ratio of DITP and Ext-TEB is 1:3. In Fig. 5e and f, the stoichiometric ratio of DCTP and Ext-TEB is 1:3.

We have added such piece of information in the inset and caption of Fig. 5.

8, Fig. S1e,f: the STM image is fuzzy, some molecules seem to be still mobile, so the model and the intermolecular interactions are hard to understand, please detail.

Author reply: The self-assembly of DBTP on Ag(111) has been reported previously (*J. Phys. Chem. C* **2013**, *117*, 302; *J. Phys. Chem. C* **2017**, *121*, 8033). As shown in Reply Fig. 6 (copy from *J. Phys. Chem. C* **2013**, *117*, 302), the self-assembly structure is the same to that shown in Supplementary Fig. 1e and f. Physically adsorbed DBTP monomers self-assemble into two-dimensional porous network, through Br \cdots Br bonds and Br \cdots H bonds (*J. Phys. Chem. C* **2013**, *117*, 302; *J. Phys. Chem. C* **2017**, *121*, 8033). Additionally, regular propeller-like patterns are observed inside the hexagonal pores. The fuzzy pattern can therefore be attributed to a caged molecule hopping around four stable adsorption configurations, facilitated by weak Br \cdots Br and Br \cdots H bond between the caged monomer and the adjacent molecules (*J. Phys. Chem. C* **2013**, *117*, 302). The corresponding structural model is provided in Reply Fig. 6c (copy from *J. Phys. Chem. C* **2013**, *117*, 302).

Reply Fig. 6 | The self-assembly of DBTP on Ag(111). **a**, STM topography images of hexagonal and rectangular networks with a regular propellerlike pattern inside the hexagons. **b**, High-resolution STM images **a**. **c**, Corresponding structural model. (copy from *J. Phys. Chem. C* **2013**, *117*, 302)

[**Editorial Note:** This figure is "Reprinted (adapted) with permission from *J. Phys. Chem. C* 2013, 117, 1, 302–306 Publication Date: December 11, 2012 <https://doi.org/10.1021/jp309554z>

Copyright © 2012 American Chemical Society. Copyright 2012 American Chemical Society."]

As suggested, we have made added the corresponding descriptions:

Added ref. 5 and 6 in the revised supporting materials.

Supplementary Information, Page S2, Line 9, added “**The deposition of 4,4’-Dibromo-p-terphenyl (DBTP, structural model is shown in the inset of Supplementary Fig. 1e) on Ag(111) held at RT leads to the formation of two-dimensional porous network, through Br···Br bonds and Br···H bonds. Regular propeller-like patterns are observed inside the hexagonal pores. This phenomenon is attributed to a caged molecule hopping around four stable adsorption configurations, facilitated by weak Br···Br and Br···H bond between the caged monomer and adjacent molecules. The detailed structural model can be seen in previous reports.^{5,6}**”

9, Fig. S13a,b: caption: Cu(111), not Ag(111).

Author reply: We have redrawn this figure (Supplementary Fig. 20 in the revised version), and the mistake in the caption has been corrected.

Reviewer #2 (Remarks to the Author):

Comments:

In the manuscript, the authors report inter-molecular radical transfer reactions with the aid of STM, TPD, XPS and DFT. Upon this, the dehydrogenation of terminal alkynes was promoted with the assistance of radicals from the dehalogenation of halobenzene molecules. In the meanwhile, the dehalogenated halobenzene molecules were passivated by detached hydrogen atoms from alkynes. In addition, such inter-molecular radical transfer mechanism was investigated in other systems, e.g., reactions of amino/imino derivatives, showing its potential universality. Despite its wide exploration in solution-phase chemistry, inter-molecular radical transfer in on-surface synthesis has been rarely reported. Therefore, this work is novel and of great significance, which offers an evidence in real space at atomic precision for the classical inter-molecular radical transfer.

Author reply: We thank the reviewer for the careful reviewing and the positive evaluation.

However, some data interpretation and the universality of the intermolecular radical transfer lack strong evidences. More discussions and experimental proofs should be given to further confirm them. A few suggestions are listed below. Once these concerns are satisfactorily addressed, the paper could be recommended for the publication in Nat. Commun.

Author reply: In the following, all the raised questions are addressed one by one in details:

1, In Fig. S5 and S6, the authors showed “the dehydrogenation of Ext-TEB occurs after annealing the sample at 380 K/400 K”. What are the products in each case? Alkynyl-Ag-Alkynyl linked organometallic motifs or covalent structures by Glaser coupling? The former would indicate that Br adatoms also have catalytic performance on the dehydrogenation of terminal alkyne.

Author Reply: We thank the reviewer for the excellent comments.

Reply Fig. 7a gives a representative STM image after annealing the Ext-TEB sample with the presence of Br adatoms to 400 K. We did not observe alkynyl-Ag-alkynyl linkages. Instead, enyne products are formed, similar to what occurs without the presence of aryl halide precursors (ACS Nano **2021**, 15, 4937; J. Am. Chem. Soc. **2021**, 143, 9461; Nat. Chem. **2016**,

8, 678; *Chem. Commun.* **2020**, 56, 8659) (the detailed reasons for forming enyne products can be seen in the reply of question (II) of referee 1). This result agrees well with the large reaction barrier (Supplementary Fig. 9e).

Reply Fig. 7c gives a representative STM image after annealing the Ext-TEB sample with the presence of Ag hybrid species at 380 K, in which we did observe alkyne-Ag-alkynyl products. This suggests that Ag hybrid species partially triggered the dehydrogenation reaction of alkyne groups. Though the reaction temperature (380 K) is lower than that for the coupling reaction of isolated alkynes (400 K), it is consistently higher than that of the radical transfer process (340 K). This conclusion agrees well with the DFT calculations that the dehydrogenation reaction barrier is 1.29 eV with the affinity of Ag hybrid species (Supplementary Fig. 10e), higher than that for the radical transfer mechanism (1.05 eV in Fig. 3).

Reply Fig. 7 | The reaction behavior of Ext-TEB with the presence of Br adatoms and Ag hybrid species, respectively. a, Formation of enyne products after annealing the Ext-TEB sample with Br adatoms at 400 K. **b**, The corresponding structural model. **c**, The STM image of organometallic products obtained after annealing the Ext-TEB sample with Ag hybrid species at 380 K. **d**, Structural models of products marked in **c**. Tunneling parameters are $I_t = 100$ pA and $V_b = -1$ V for all the STM images. Image sizes are 15×15 nm² for **a** and **c**. Ag, C

and H atoms are represented by blue, gray and white circles, respectively.

As suggested, we have made the corresponding corrections:

Modify Supplementary Fig. 9 and 10, and the corresponding description in the captions.

Supplementary Information, Page S11, Line 7, added “We did not observe the alkynyl-Ag-alkynyl linkages after annealing the sample at 400 K, as shown in Supplementary Fig. 9c. Instead, enyne products are formed, which is same to that without the presence of aryl halide precursors.¹⁻⁴”

Supplementary Information, Page S11, Line 12, modification “As shown in Supplementary Fig. 9e, the presence of Br adatom does not facilitate the dehydrogenation of terminal alkyne, resulting in a high energy barrier of 2.13 eV.

The high reaction temperature, the formation of enyne products and the calculated high reaction barrier suggest that the possibility of employing Br adatom as the active center can be ruled out.”

Supplementary Information, Page S13, Line 6, added “With the participation of Ag hybrid species, the dehydrogenation of terminal alkyne occurs after annealing the Ext-TEB sample at 380 K, which is 40 K higher than that in the bi-component system (Fig. 2). As shown in Supplementary Fig. 10c and d, the dehydrogenated Ext-TEB molecules connect with each other or with DBTP monomer by alkynyl-Ag-alkynyl and alkynyl-Ag-phenyl species.”

Supplementary Information, Page S13, Line 16, modification “The control experiments and the DFT calculations suggest that Ag hybrid species do reduce the dehydrogenation barrier of alkynes, though they are not the active center for the reaction taken place at 340 K in the bi-component systems shown in Fig. 2.”

2, The author highlighted a lot of times that “Debrominated phenyl radicals are passivated by H atoms with 100% reaction selectivity”, which is indeed the key point of the reaction mechanism of inter-molecular radical transfer. However, the conventional STM images do not have enough resolution to identify the H saturation. Nc-AFM with a CO-functionalized probe will help.

Author reply: We agree with the referee that the hydrogen passivation is indeed the key point

to support the inter-molecular radical transfer mechanism. We therefore confirmed the hydrogen passivation by complementary control experiments:

1. We further annealed the sample shown in Supplementary Fig. 4 at 420 K (Reply Fig. 8), and observe that the self-assembly islands remain unaltered (annealing at 370 K: $a = 1.72 \pm 0.03$ nm, $b = 1.01 \pm 0.02$ nm, $\theta = 150 \pm 1^\circ$; annealing at 420 K: $a = 1.69 \pm 0.03$ nm, $b = 1.01 \pm 0.02$ nm, $\theta = 150 \pm 1^\circ$). If the monomers were not passivated, Ullmann-type coupling would have occurred at the elevated temperatures.

2. Previous reports have demonstrated that the introduction of phenol derivatives can effectively transform the metal-organic oligomers to hydrogen-passivated monomers through the breakage of metal-organic bonds (*Chem. J. Chinese Universities*, **2021**, 42, 1241). In this control experiments, we choose 1,3,5-tris(4-hydroxyphenyl)benzene (THPB) as the phenol derivative. Reply Fig. 9a shows that DBTP molecules form one-dimensional organometallic chains via Ph-Ag-Ph bonds on Ag(111) after annealing at 400 K. By subsequently depositing THPB molecules on the surface held at 400 K, organometallic wires gradually decompose, resulting in the formation of close-packed assembly islands, as shown in Reply Fig. 9b. According to the reference (*Chem. J. Chinese Universities*, **2021**, 42, 1241), monomers in this assembly island are H-passivated DBTP (the reaction path is schematically illustrated in Reply Fig. 9c). The self-assembled structure ($a = 1.73 \pm 0.03$ nm, $b = 1.01 \pm 0.02$ nm, $\theta = 150 \pm 1^\circ$) shown in Reply Fig. 9b is same to that shown in the Supplementary Fig. 4a and Reply Fig. 8.

We therefore believe it is safe to conclude that monomers shown in Supplementary Fig. 4a are passivated (by hydrogen atoms) DBTP monomers.

Reply Fig. 8 | Thermal annealing for the sample (Ext-TEB + DBTP)/Ag(111). a and b

Representative STM images after annealing the (Ext-TEB + DBTP)/Ag(111) sample at 370 K and 420 K, respectively. Unit cells are superposed. Tunneling parameters are $I_t = 100$ pA and $V_b = -1$ V for all the STM images. Image sizes are 10×10 nm² for **a** and **b**.

Reply Fig. 9 | Structural evolution of DBTP with introducing THPB molecule. **a**, STM topographic image after depositing DBTP on Ag(111) held at 400 K. **b**, STM topographic image after depositing THPB on the surface shown in **a** held at 400 K. **c**, Scheme of the reaction pathway. Tunneling parameters are $I_t = 100$ pA and $V_b = -1$ V for all the STM images. Image sizes are 10×10 nm² for **a** and **b**. Ag, C, O, Br and H atoms are represented by silver, gray, red, brown and white circles, respectively.

As the passivation of radicals can be confirmed by control experiments (Reply Fig. 8 and 9), we did not perform additional nc-AFM investigations.

We added the corresponding descriptions in the revised version:

Added ref. 10 in the revised supporting materials.

Added Supplementary Fig. 5 and 6.

Page 5, Line 21, added “The passivation of phenyl radicals by hydrogen atoms is further confirmed via control experiments (Supplementary Fig. 5 and 6).”

Supplementary Information, Page S7, Line 1, added “**5. Control experiments to confirm H-passivated DBTP**

The hydrogen passivation is the key point to support the inter-molecular radical transfer mechanism. We therefore confirmed the hydrogen passivation by complementary control experiments:

1. We further annealed the phase shown in Supplementary Fig. 4 at 420 K (Supplementary Fig. 5), and observe that the self-assembly islands stayed unaffected (annealing at 370 K: $a = 1.72 \pm 0.03$ nm, $b = 1.01 \pm 0.02$ nm, $\theta = 150 \pm 1^\circ$; annealing at 420 K: $a = 1.69 \pm 0.03$ nm, $b = 1.01 \pm 0.02$ nm, $\theta = 150 \pm 1^\circ$). If the monomers were not passivated, Ullmann-type reactions would have taken place at the elevated temperatures.

2. Previous reports demonstrated that the introduction of phenol derivatives can effectively transform the metal-organic oligomers to hydrogen-passivated monomers through the breakage of metal-organic bonds¹⁰. In this control experiments, we choose 1,3,5-tris(4-hydroxyphenyl)benzene (THPB) as the phenol derivative. Supplementary Fig. 6a shows that DBTP molecules form one-dimensional organometallic chains via Ph-Ag-Ph bonds on Ag(111) after annealing at 400 K. By subsequently depositing THPB molecules on the surface held at 400 K, organometallic wires gradually decompose, resulting in the formation of close-packed assembly islands, as shown in Supplementary Fig. 6b. According to the reference¹⁰, monomers in this assembly island are H-passivated DBTP (the reaction path is schematically illustrated in Supplementary Fig. 6c). The self-assemble structure ($a = 1.73 \pm 0.03$ nm, $b = 1.01 \pm 0.02$ nm, $\theta = 150 \pm 1^\circ$) shown in Supplementary Fig. 6b is same to that shown in the Supplementary Fig. 4 and 5.

We therefore believe it is safe to conclude that monomers shown in Supplementary Fig. 4a are passivated (by hydrogen atoms) DBTP monomers.”

3, According to the inter-molecular radical transfer mechanism proposed by the authors, the terminal alkyne should be very close to the dibrominated halobenzene to make the reaction

occur. This implies that if the molecular coverage is extremely low or extremely high the reaction yield should be lower because the chance that two reactants meet together is reduced. Did the authors perform these experiments?

Author reply: We thank the reviewer for this comment. The diffusion of precursors is indeed important, as it determines whether the initial states of the DFT calculation shown in Fig. 3 can be achieved.

As described in the reply to question 2a of referee 1, both precursors diffuse sufficiently on Ag(111). Experiments shown in Fig. 2 are performed with the total coverage close to 1 ML (at low coverage, the pristine monomers diffuse significantly, as evidenced by the fuzzy images shown in Reply Fig. 1). Experiments with extremely low coverage were not conducted. We did not observe a decrease of reaction yield at high coverage (Actually, the yield at high coverage is close to 100%, as shown in Fig. 2 in the manuscript).

On the other hand, we found that the reaction products are mainly influenced by the stoichiometric ratio of precursor molecules, as shown in Reply Fig. 10.

Reply Fig. 10 | Reaction behavior after co-depositing DBTP and Ext-TEB molecules with different stoichiometric ratios on Ag(111). a and b, excess of aryl bromides; c and d,

comparable for both groups; **e** and **f**, excess of alkyne groups. Tunneling parameters are $I_t = 100$ pA and $V_b = -1$ V for all the STM images. Image sizes are 20×20 nm² for all the STM images. Ag, C and H atoms are represented by silver, gray and white circles, respectively.

4, As for the universality, the reviewer suggests the authors also use a small molecule like CH₃CH₂Br as the radical source, which has a sufficient mobility on the surfaces. According to the mechanism proposed by the authors, it could also promote the dehydrogenation of terminal alkynes. On the contrary, if a halobenzene molecule, as the radical source, is too large, it may not promote the dehydrogenation of terminal alkynes because of the limited diffusion on surfaces. The author should consider to accomplish these control experiments to strength the conclusion.

Author reply: Thanks for the suggestions. We did not perform extra experiments with smaller bromo-substitute aromatics precursors because:

Firstly, the yield is already close to 100% in Fig. 2 (almost all the alkyne groups have hydrogen detached and form alkynyl-Ag-alkynyl motifs, and almost all the phenyl radicals are passivated by hydrogen atoms) by choosing DBTP as bromo-substitute aromatics precursors.

Secondly, the reason to choose smaller precursor molecules is to increase the mean free path (MFP) of the monomers. Actually, DBTP monomers diffuse sufficiently on Ag(111) at the reaction temperature (see reply of question 2a of referee 1 and Reply Fig. 1 for details).

Thirdly, in this paper, we systematically studied the influence of phenyl radicals on the hydrogen transfer reaction. It is a very clever idea to extend from phenyl radicals to sp³-hybridized C-centered radicals by choosing e.g. CH₃CH₂Br as precursor molecules. We will conduct such studies in future work.

5, The bibliography should be expanded, because plenty of similar studies on radical promoted dehydrogenation of terminal alkynes have been reported, though without direct evidences, e.g. ACS Nano 2021, 15, 3, 4937-4946; Phys. Chem. Chem. Phys. 2018, 20, 11081-11088. Angew. Chem. Int. Ed. 2017, 56, 4762-4766. These references could be in turn used to support the conclusion of this work.

Author reply: Thank the reviewer for the comment. As suggested, we have added those

references in the revise version: ref. 23, 54, 55, 56 and 57.

Minor comments:

1, Please check the English grammar in the manuscript carefully.

Author reply: We thank the referee for the careful review. As suggested, we went through the manuscript carefully and polished the language in the revised version.

2, Fig. 2d. some bright dots filling in the self-assembled island should be Ag adatoms, as seen in ACS Nano 2018, 12, 2267-2274; Chem. Eur. J. 2017, 23, 6190-6197. Therefore, the author should be more careful measuring the length of a monomer.

Author reply: We agree with the referee that debrominations of DBTP in Fig. 2 should be examined more carefully. Br adatoms interact with benzenes via the C-H \cdots Br interaction (*J. Phys. Chem. Lett.* **2017**, 8, 326; *J. Phys. Chem. C* **2018**, 122, 25681). In contrast, Ag adatoms interact with aryl bromides through the Ph-Br \cdots Ag interaction (*ACS Nano* **2018**, 12, 2267). In both situations, the distance between the adatom and the interacting phenyl ring is much longer than that between the Br atom and the adjacent phenyl ring within a pristine DBTP monomer. It is therefore possible to determine precisely, whether Br atoms are detached by means of distance measurements.

As shown in Reply Fig. 11b-d, two distances are measured. The protrusion closer to phenyl groups (0.42 ± 0.03 nm) can be ascribed to Br atoms covalently bond to phenyls. Protrusion further away from phenyl groups (0.59 ± 0.05 nm) can be assigned to Ag adatoms or detached Br adatoms. Consequently, the monomers shown in Reply Fig. 11b-d can be attributed to the pristine monomer, monomer with one Br atom detached, and monomer with Br atoms at both sides detached, respectively.

Reply Fig. 11 | Distance measurements. a, Self-assembly of DBTP after annealing the sample at 340 K. **b-d**, High resolution STM images of three representative reaction products of DBTP. The corresponding line profiles, measured distances and structural models are shown in the lower panels. Tunneling parameters are $I_t = 100$ pA and $V_b = -1$ V for all the STM images. The image size is 10×10 nm² for **a**. Br, C, and H atoms are represented by brown, gray, and white circles, respectively.

We have added the corresponding descriptions in the revised version of the manuscript:

Added ref. 7-9 in the revised supporting materials.

Added Supplementary Fig. 3.

Page 5, Line 17, added “Further detailed information on the Br abstraction are presented in Supplementary Fig. 3.”

Supplementary Information, Page S5, Line 1, added “**3. Distance measurements for DBTP molecule and its debrominated products**

According to previous reports, Br adatoms interact with benzenes via the C-H...Br interaction,^{7,8} and Ag adatoms interact with aryl bromides through the Ph-Br...Ag interaction.⁹ In both situations, the distance between the adatom and the interacting phenyl ring is much longer than that between Br atom and the adjacent phenyl within a pristine DBTP monomer. It is therefore possible to determine precisely, whether Br atoms are detached by means of distance measurements.

As shown in Supplementary Fig. 3b-d, two distances are measured. The protrusion closer to phenyl groups (0.42 ± 0.1 nm) can be ascribed to Br atoms covalently bond to phenyls. Protrusion further away from phenyl groups (0.59 ± 0.2 nm) can be assigned to Ag adatoms or detached Br adatoms. Consequently, the monomers shown in Supplementary Fig. 3b-d can be attributed to the pristine monomer, monomer with one Br atom detached, and monomer with Br atoms at both sides detached, respectively.”

3, “significant desorption is observed for both the alkynyl-Ag species and passivated phenylene”. Is it possible to desorb a alkynyl-Ag species from a Ag(111) surface?

Author reply: According to our experimental observations, alkynyl-Ag species indeed

desorbed from Ag(111) after annealing the sample to 450 K. The evidence is provided in the following:

1. As shown in Reply Fig. 12, after further annealing the sample shown in Fig. 2 at 450 K, alkynyl-Ag species desorb significantly. Meanwhile, the coverage of DBTP also decreases, and the remaining molecules aggregate into large-scale islands (Reply Fig. 12a).

2. In TP-XPS spectra of the (DBTP + Ext-TEB)/Ag(111) sample, the intensity of C 1s core levels for the bi-component system decreases dramatically at temperatures higher than 450 K (Supplementary Fig. 8), signaling the significant desorption of molecules on Ag(111) at 450 K.

Reply Fig. 12 | Desorption of alkynyl-Ag species and H-passivated DBTP molecules after annealing. **a** Large-scale and **b** zoomed STM images after annealing the DBTP + Ext-TEB/Ag(111) sample at 450 K. Tunneling parameters are $I_t = 100$ pA and $V_b = -1$ V for all the STM images. Image sizes are 100×100 nm² for **a** and 30×30 nm² for **b**.

We added the corresponding descriptions in the revised version:

Added Supplementary Fig. 7.

Page 5, Line 26, added “see Supplementary Fig. 7 for details”.

Supplementary Information, Page S9, Line 1, added “**6. Desorption of reaction products after annealing.**”

As shown in Supplementary Fig. 7, after further annealing the sample shown in Fig. 2 at 450 K, alkynyl-Ag species desorb significantly. Meanwhile, the coverage of DBTP also decreases, and the remaining molecules aggregate into large-scale islands.”

4, Page 3. “(2) Intermolecular radical transfer should be favored energetically, with the energy barrier of the radical transfer being lower than or comparable to that of the radical generation.” This would only determine the radical transfer is not the rate-limiting step, instead of implying an intermolecular radical transfer cannot happen.

Author reply: We agree with the referee. We mentioned “(2) Intermolecular radical transfer should be favored energetically, with the energy barrier of the radical transfer being lower than or comparable to that of the radical generation.” because the self-passivation of phenyl radicals may precede the radical transfer process if the radical transfer barrier is much higher than the radical generation barrier. This sentence is deleted in the revised manuscript as we notice it describes the same thing as reason (3).

Page 3, Line 18, Delete “Inter-molecular radical transfer should be favored energetically, with the energy barrier of the radical transfer being lower than or comparable to that of the radical generation.”

Reviewer #3 (Remarks to the Author):

Comments:

The field of on-surface-synthesis has been highly active within experimental surface science for the past 1.5 decades, using a combination of molecular resolution imaging techniques, chemical spectroscopy and theoretical modelling to address how organic molecules adsorbed on solid surfaces may react to form covalently bonded organized structures.

Here, the authors add significant new insights into possible principles of on-surface-synthesis by proposing a new radical transfer mechanism to facilitate dehydrogenation and molecular coupling. Their experiments involve co-adsorption of a three-spoke terminal alkyne with a tri-phenyl halide on a surface of silver. Upon heating the substrate, they find the co-adsorbed three-spoke alkynes to react differently than when adsorbed alone, and form an extended network attributed to dehydrogenated three-spoke alkynes interconnected by coordinating to Ag atoms (as observed by molecular level scanning tunneling microscopy). The authors attribute this unexpected finding to a radical transfer mechanism where dissociation of the halogen (Br) atom from the tri-phenyl halide, creates a reactive tri-phenyl radical which abstracts hydrogen from the terminal alkyne, leading to the coordination complex. This proposed novel reaction mechanism is corroborated by DFT calculations of the reaction path as well as an impressive array of control and support experiments which all form a consistent picture as far as this referee can judge. The paper builds on a large amount of previous knowledge for the on-surface reaction mechanisms of alkynes (Glaser coupling) and phenyl halides (Ullmann coupling) which makes it somewhat challenging to read, but the authors do a good job of referring the existing literature to make the story tractable.

The evidence for the creation of the intermediate phenyl radical and the eventual claimed saturation of this radical by the hydrogen abstracted from the terminal alkyne is overall circumstantial, but well supported by the consistent picture leading to plausible explanations for all the observed phenomena, e.g for the observed threshold temperatures for radical transfer reactions (or lack thereof) for I, Br and Cl based tri-phenyl halides with different activation energies for reaction. My only question concerns the control experiments reported in Fig. 4 using a stoichiometric excess of the three-spoke alkyne:

What happens if the reaction is instead performed with an excess of the tri-phenyls? Will one in this situation observe molecules created by ph-ph coupling since there is then not enough hydrogen abstracted from the available alkynes to passivate all tri-phenyl radicals? Discussion of such data would provide further support for the proposed reaction mechanism.

Overall, I find the paper should be of interest to organic chemists in general as it reports new insights into radical transfer reactions for molecules adsorbed on metal surfaces (where the interaction of the radical with the surface electronic structure has been thought to suppress radical transfer reactions). The paper is particularly interesting for readers in the field of molecular nanoscience, since the presented data for other (amine-containing) compounds suggest the radical transfer mechanism induced by co-adsorption with a phenyl-halide might be a quite general avenue to facilitate hydrogen abstraction, opening up for new pathways within on-surface-synthesis.

Author reply: We thank the reviewer for the thoughtful and positive evaluations.

My only question concerns the control experiments reported in Fig. 4 using a stoichiometric excess of the three-spoke alkyne: What happens if the reaction is instead performed with an excess of the tri-phenyls? Will one in this situation observe molecules created by phe-phe coupling since there is then not enough hydrogen abstracted from the available alkynes to passivate all tri-phenyl radicals? Discussion of such data would provide further support for the proposed reaction mechanism.

Author reply: We thank the reviewer for the comments.

As suggested, we performed control experiments with an excess of DBTP molecules in the bi-component system (see Reply Fig. 13), and added the corresponding descriptions in the revised version:

Added **Supplementary Fig. 14** (same to that shown in reply Fig. 13).

Page 8, Line 21, added “**On the other hand, when introducing an excess of DBTP, all dehydrogenated alkynyl groups interact with phenyl groups and adatoms, expressing phenyl-Ag-alkynyl bonds. As DBTP molecules are in excess, phenyl-Ag-phenyl motifs are also observed (see Supplementary Fig. 14 for details).**”

Supplementary Information, Page S19, Line 1, added “**13. Reaction behavior after depositing an excess of DBTP molecules in the bi-component system.**

Supplementary Fig. 14a gives a representative STM image after co-depositing Ext-TEB and an excess of DBTP molecules on Ag(111) held at RT. The assembled structure of precursor molecules is same to that observed in Fig. 2a. Annealing the sample at 370 K leads to the formation of various oligomers, as shown in Supplementary Fig. 14b and c. All dehydrogenated alkynyl groups interact with phenyl groups via phenyl-Ag-alkynyl bonds due to the excess of DBTP. Meanwhile, phenyl-Ag-phenyl motifs are also observed.”

Reply Fig. 13 | Reaction behavior with an excess of DBTP. **a**, Large-scale STM image after co-deposition of Ext-TEB and DBTP on Ag(111) held at RT (DBTP is excess). **b** and **c**, STM topographic images obtained by annealing the sample from RT to 370 K. The corresponding structural models of products shown in the right panels. Tunneling parameters are $I_t = 100$ pA and $V_b = -1$ V for all the STM images. Image sizes are 50×50 nm² for **a** and 20×20 nm² for **b** and **c**. Ag, C and H atoms are represented by blue, gray and white, respectively.

REVIEWERS' COMMENTS

Reviewer #1 (Remarks to the Author):

The authors made a long and exhaustive reply to all the reviewers' remarks and criticisms, after having performed several additional experiments and simulations. All this new information drastically improves the scope and quality of the presented results, I now favorably support the publication.

I have only one request concerning the reply #6 to reviewer #1 about the apparent excess of Br adatoms. I fully agree this known phenomenon is still unclear, but, and especially for this reason, the small discussion presented is interesting and would be worth being added as a paragraph in the supporting information.

Reviewer #2 (Remarks to the Author):

The editors have satisfactorily addressed my previous concerns and have now provided an improved manuscript that I consider adequate for publication in Nat. Commun. as it is.

Reviewer #3 (Remarks to the Author):

I have read the revised manuscript with interest and find it has improved significantly with the addition of further experimental data and calculations. In particular, the evidence for hydrogen passivation of the debrominated terphenyl has been strengthened by further control experiments. The one question in my original report regarding experiments with a stoichiometric excess of DBTP have been adequately addressed with the new SI figure 14.

Reviewers' comments:

Reviewer #1 (Remarks to the Author):

Comments:

The authors made a long and exhaustive reply to all the reviewers' remarks and criticisms, after having performed several additional experiments and simulations. All this new information drastically improves the scope and quality of the presented results, I now favorably support the publication.

I have only one request concerning the reply #6 to reviewer #1 about the apparent excess of Br adatoms. I fully agree this known phenomenon is still unclear, but, and especially for this reason, the small discussion presented is interesting and would be worth being added as a paragraph in the supporting information.

Author reply: We thank the reviewer for the careful reviewing and the positive evaluation.

As suggested, we added the corresponding descriptions in the revised version:

Page 8, Line 15, added “**the reason why plenty of Br adatoms are observed can be seen in Supplementary discussion**”

Supporting Materials, Page S31, Line 1, added “**Suppl. Discussion 1. Explanations for the high number of Br adatoms in STM images**

The observation of the excess of Br adatoms is rather common in Ullmann reactions. However, previous papers did not provide definitive explanations for such phenomenon.²⁶⁻²⁸

In our case, the desorption of molecules in the bi-component system at the reaction temperature is not obvious according to our TP-XPS spectra. Moreover, the adsorption of Br atoms generated in the organic evaporator can be ruled out, as no Br adatoms are detected by solely depositing DBTP on Ag(111) held at RT (see Supplementary Fig. 1e). We therefore believe such phenomenon may arise from one or several of the following reasons: (1) there may exist the self-assembly islands of passivated DBTP without the decoration of Br adatoms, which we did not observe. (2) Plenty of monomers adsorb along the step edges. (3) Small amount of passivated bromobenzene molecules desorb from surface, and the desorption amount is lower than the detection limit of TP-XPS.”

Added **reference 26-28** in the supporting information.

Reviewer #2 (Remarks to the Author):

The editors have satisfactorily addressed my previous concerns and have now provided an improved manuscript that I consider adequate for publication in Nat. Commun. as it is.

Author reply: We thank the reviewer for the thoughtful and positive evaluations.

Reviewer #3 (Remarks to the Author):

I have read the revised manuscript with interest and find it has improved significantly with the addition of further experimental data and calculations. In particular, the evidence for hydrogen passivation of the debrominated terphenyl has been strengthened by further control experiments. The one question in my original report regarding experiments with a stoichiometric excess of DBTP have been adequately addressed with the new SI figure 14.

Author reply: We thank the reviewer for the careful reviewing and the positive evaluation.